# A cellular and regulatory map of the GABAergic nervous system of *C. elegans*

Marie Gendrel*, Emily G Atlas, Oliver Hobert*

Department of Biological Sciences, Howard Hughes Medical Institute, Columbia University, New York, United States

**Abstract** Neurotransmitter maps are important complements to anatomical maps and represent an invaluable resource to understand nervous system function and development. We report here a comprehensive map of neurons in the *C. elegans* nervous system that contain the neurotransmitter GABA, revealing twice as many GABA-positive neuron classes as previously reported. We define previously unknown glia-like cells that take up GABA, as well as 'GABA uptake neurons' which do not synthesize GABA but take it up from the extracellular environment, and we map the expression of previously uncharacterized ionotropic GABA receptors. We use the map of GABA-positive neurons for a comprehensive analysis of transcriptional regulators that define the GABA phenotype. We synthesize our findings of specification of GABAergic neurons with previous reports on the specification of glutamatergic and cholinergic neurons into a nervous system-wide regulatory map which defines neurotransmitter specification mechanisms for more than half of all neuron classes in *C. elegans*.

## Introduction

Since the days of Ramón y Cajal, the generation of maps of the brain constitutes a central pursuit in the neurosciences. The nervous system of the nematode *C. elegans* constitutes the currently best mapped nervous system. Available *C. elegans* nervous system maps include a lineage map of all neurons (*Sulston, 1983*) and an anatomical map that describes all individual neuron types not just in terms of overall morphology but also synaptic connectivity (*Jarrell et al., 2012*; *White et al., 1986*). One type of map that complements anatomical maps and that is critical to understand neuronal communication is a map that assigns a neurotransmitter identity to all neurons in the nervous system. Comprehensive maps of modulatory, monoaminergic neurons (e.g. serotonergic, dopaminergic) have been known for some time in *C. elegans* (*Chase and Koelle, 2007*), but comprehensive maps of the most prominent small molecule neurotransmitter systems employed throughout all animal nervous systems – glutamate (Glu), acetylcholine (ACh) and GABA – are only now emerging. We have recently defined the complete set of glutamatergic (*Serrano-Saiz et al., 2013*) and cholinergic neurons in *C. elegans* (*Pereira et al., 2015*) and in this third neurotransmitter-mapping paper, we describe our analysis of GABA-positive neurons, expanding previous work that had begun to define GABAergic neurons in *C. elegans* (*McIntire et al., 1993b*).

GABA is a neurotransmitter that is broadly used throughout all vertebrate and invertebrate nervous systems. In vertebrates, GABA is used as a neurotransmitter by many distinct neuron types throughout the CNS (30–40% of all CNS synapses are thought to be GABAergic; [*Docherty et al., 1985*]) and alterations of GABAergic neurotransmission are the cause of a number of neurological diseases in humans (*Webster, 2001*). One intriguing issue, unresolved in vertebrates due to the complexity of their nervous systems, is the cellular source of GABA and the fate of GABA after cellular release. The expression of the biosynthetic enzyme for GABA, glutamic acid decarboxylase (GAD), defines neurons that have the capacity to synthesize GABA, but the existence of plasma

*For correspondence: mg3088@columbia.edu (MG); or38@columbia.edu (OH)

membrane transporters for GABA (called GAT) indicates that GABA can also be 'acquired' by neurons via transport and not synthesis (*Zhou and Danbolt, 2013*). Does GABA uptake merely occur to clear GABA, thereby controlling the duration of a GABAergic signal, or do cells take up GABA to then reemploy it, e.g. by using vesicular GABA transporters (VGAT) to synaptically release GABA? Also, is GABA only taken up by neurons that are innervated by GABA neurons? A precise map of GAD-, GAT- and VGAT-expressing neurons with single neuron resolution would shed light on these issues, but has not yet been produced in vertebrate nervous systems. In this resource paper, we provide such a map in the nematode *C. elegans.*

Previous studies have ascribed a GABAergic neurotransmitter identity to 26 *C. elegans* neurons, which fall into six anatomically and functionally diverse neuron classes. These numbers amount to less than 10% of all neurons (302 hermaphroditic neurons) and neuron classes (118 anatomically defined neuron classes) (*McIntire et al., 1993a*, *1993b*; *Schuske et al., 2004*). Not only is this substantially less than the number of neurons that use conventional excitatory neurotransmitters (Glu: 39 classes, ACh: 52 classes; [*Pereira et al., 2015*; *Serrano-Saiz et al., 2013*]), but, given the abundance of GABAergic interneurons in vertebrates, it is also striking that only one of the previously identified GABA neurons is an interneuron (*McIntire et al., 1993b*). However, the *C. elegans* genome contains at least seven predicted ionotropic GABA receptors (*Hobert, 2013*) and at least some of them are expressed in cells that are not synaptically connected to the previously defined GABA neurons (*Beg and Jorgensen, 2003*; *Jobson et al., 2015*). We therefore suspected that additional GABAergic neurons may have been left undetected. Using a refined GABA antibody staining protocol and improved reporter gene technology, we extend here the original set of six GABA-positive neuron classes by another ten additional GABA-positive cell types, seven of them neuronal cell types.

Knowledge of the complete and diverse set of neurons sharing the expression of a specific neurotransmitter system allows one to ask how the expression of a shared identity feature is genetically programmed in distinct neuron types. As mentioned above, the usage of GABA as a neurotransmitter represents a unifying terminal identity feature for a diverse set of neurons in invertebrate and vertebrate nervous systems. Given the diversity of GABAergic neuron types, it is perhaps not surprising that no unifying theme of GABAergic identity specification has yet been discovered. Nevertheless, while distinct GABAergic neuron types use distinct transcription regulatory codes, some transcription factors appear to be repeatedly used by distinct GABAergic neuron types. For example, in vertebrates, the GATA-type transcription factors GATA2/3 are employed for GABAergic identity specification by midbrain and spinal cord neurons (*Achim et al., 2014*; *Joshi et al., 2009*; *Kala et al., 2009*; *Lahti et al., 2016*; *Yang et al., 2010*). We explore here whether the theme of reemployment of a transcription factor in different GABAergic neurons also exists in *C. elegans.*

Another question that pertains to the development of GABAergic neurons relates to the stage at which regulatory factors act to specify GABAergic neurons. Previous studies on the specification of vertebrate GABAergic neurons have so far uncovered factors that act at distinct stages of GABAergic neuron development (*Achim et al., 2014*). However, there is still a remarkable dearth of knowledge about transcriptional regulators that are expressed throughout the life of GABAergic neurons to not only initiate but also maintain the differentiated state of GABAergic neurons. Such type of late acting transcriptional regulators have previously been called 'terminal selectors' and these terminal selectors have been identified to control the identity of distinct neuron types utilizing a variety of distinct neurotransmitter systems (*Hobert, 2008*, *2016a*).

In *C. elegans*, previous work has shown that the Pitx2-type transcription factor *unc-30* selectively specifies the identity of D-type motor neurons along the ventral nerve cord (*Cinar et al., 2005*; *Eastman et al., 1999*; *Jin et al., 1994*). As a terminal selector of D-type motor neuron identity, UNC-30 protein directly controls the expression of GABA pathway genes (*Eastman et al., 1999*) as well as a plethora of other D-type motor neuron features (*Cinar et al., 2005*), including their synaptic connectivity (*Howell et al., 2015*). Since *unc-30* is expressed in non-GABAergic neurons (*Jin et al., 1994*), *unc-30* is not sufficient to induce the GABAergic neuronal identity, possibly because *unc-30* may act with as yet unknown cofactors in the D-type motor neurons. We identify here a putative cofactor for *unc-30* in the form of the *elt-1* gene, the *C. elegans* ortholog of the vertebrate Gata2/3 transcription factors, which specify GABAergic neurons in vertebrates (*Achim et al., 2014*).

The acquisition of GABAergic identity of *C. elegans* neurons other than the D-type GABAergic motor neurons was less well understood. The RIS, AVL and DVB neurons display differentiation

defects in animals lacking the *lim-6* homeobox gene, the *C. elegans* ortholog of the vertebrate *Lmx1* LIM homeobox gene (*Hobert et al., 1999*; *Tsalik et al., 2003*) and the RME neurons display differentiation defects in animals that either carry a mutation in the *nhr-67* orphan nuclear hormone receptor, the *C. elegans* ortholog of vertebrate Tlx/NR2E1 gene and the fly gene Tailless (*Sarin et al., 2009*) or in animals lacking *ceh-10,* the *C. elegans* ortholog of the vertebrate *Vsx/Chx10* Prd-type homeobox gene (*Forrester et al., 1998*; *Huang et al., 2004*). However, the extent of the differentiation defects in these distinct mutant backgrounds has not been examined in detail. We show here that *nhr-67* controls all GABAergic identity features in the AVL, RIS and RMEL/R and RMED/V motor neurons, where it collaborates with distinct homeobox genes, *lim-6* in AVL and RIS, *ceh-10* in RMED and a novel transcription factor, *tab-1*, the *C. elegans* ortholog of vertebrate Bsx, in RMEL/R. We identify additional homeobox genes that control the identity of GABAergic neurons that we newly identify here. Taken together, our systems-wide analysis of GABA-positive cells in *C. elegans* identifies a number of distinct GABA-positive cell types that acquire and utilize GABA via diverse mechanisms and provides an extensive picture of the specification of this critical class of neurons.

## Results

### Identifying GABAergic neurons in the *C. elegans* nervous system

The previously reported set of GABAergic neurons (six neuron classes: RME, AVL, RIS, DVB, DD, VD; *Table 1*) were defined by anti-GABA antibody staining and expression of reporter transgenes that monitor expression of genes that encode the GABA biosynthetic enzyme glutamic acid decarboxylase (GAD/UNC-25) and the vesicular GABA transporter (VGAT/UNC-47) (*Jin et al., 1999*; *McIntire et al., 1993b*, *1997*). Using a modified GABA staining protocol, we observed the presence of GABA in the six previously described GABAergic neuron classes RME, AVL, RIS, VD, DD and DVB, but also detected staining in an additional set of seven neuronal cell types (RIB, SMDD/V, AVB, AVA, AVJ, ALA, AVF; *Table 1*, *Figure 1*). The identity of these GABA-positive neurons was confirmed by GABA-staining of transgenic animals that express cell-type specific markers (data not shown). Staining of these newly identified GABA-positive cells is generally weaker than in the previously identified GABA neurons (*Figure 1A,B*). In the vertebrate CNS, distinct GABAergic neuron types also show different levels of anti-GABA staining (J. Huang, pers.comm.). Anti-GABA staining in all cells, including the newly identified cells, is completely abolished in animals lacking the *unc-25* gene which codes GABA-synthesizing enzyme glutamic acid decarboxylase, thereby corroborating that the staining indeed reports on the presence of GABA (*Figure 1C*). The same pattern of staining is observed in early larvae and adult worm, with the sole exception of the AVJ neuron pair which stains more strongly in larval compared to adult stages.

### GABA uptake neurons

Neurons that package a specific neurotransmitter and release it to signal to downstream neurons do not necessarily synthesize this neurotransmitter, but may rather internalize it from their environment via neurotransmitter-specific uptake systems. For example, spinal cord motor neurons in the rat do not synthesize GABA but take up GABA after its release from presynaptic neurons (*Snow et al., 1992*). GABA synthesis is usually examined by analyzing the expression pattern of the GABA-synthesizing enzyme glutamic acid decarboxylase (GAD), encoded by the *unc-25* gene in *C. elegans*. A previously published reporter transgene for the *unc-25* locus shows expression in the set of previously identified GABA neurons listed in *Table 1* (*Jin et al., 1999*). We also detect expression of this previously described reporter in the newly identified GABA-positive RIB interneuron pair but not in any of the other newly identified GABA-positive neurons (data not shown). To test the possibility that the lack of expression in other GABA-positive neurons may be due to missing regulatory elements in the previously described reporter, we inserted a *SL2::gfp* cassette into the endogenous *unc-25* locus using the CRISPR/Cas9 system (*Figure 2B*). We observed expression in the same set of neurons as the previously described reporter transgene (*Figure 2D*; more sensitive GFP antibody staining did not reveal additional staining; data not shown). This corroborates the newly identified GABAergic identity of the RIB neuron class, but raises the question how the other newly identified GABA-positive neurons acquire GABA. To further explore this issue, we performed three sets of experiments: (a) we stained, with anti-GABA antibodies, animals that are unable to release GABA because they

**Table 1.** GABA-positive cells.

| Cells | | | anti-GABA | anti-GABA in *snf-11(-)* | anti-GABA in *unc-47(-)* | *unc-25* knock-in reporter allele | *unc-47* fosmid reporter | *unc-46* fosmid reporter | *snf-11* fosmid reporter | other fast neuro-transmitter |
|---|---|---|---|---|---|---|---|---|---|---|
| Previously identified (*McIntire et al., 1993b*) | Neuronal | RME | +++ | +++ | +++ | +++ | +++ | +++ | +++ | none |
| | | RIS | +++ | +++ | +++ | +++ | +++ | +++ | - | none |
| | | AVL | +++ | +++ | +++ | +++ | +++ | +++ | ++ | none |
| | | DVB | +++ | +++ | +++ | +++ | +++ | +++ | - | none |
| | | DD1-6 | +++ | +++ | +++ | +++ | +++ | +++ | - | none |
| | | VD1-13 | +++ | +++ | +++ | +++ | +++ | +++ | - | none |
| Newly identified | Neuronal | RIB | ++ | ++ | ++ | ++ | +++ | ++ | + | ACh (weak) |
| | | SMDD/V | +/- | + | + | - | + | - | - | ACh |
| | | AVB | ++ | + | + | - | - | + | - | ACh |
| | | AVA | +/- | +/- | + | - | - | - | - | ACh |
| | | AVJ | ++ * | + | + | - | - | - | - | none |
| | | ALA (recycling) | ++ | - | - | - | + | - | +++ | none |
| | | AVF (clearance) | + | - | - | - | - | - | +++ | none |
| | Non-Neuronal | GLR (non-neuronal) | + | - | ++ | - | - | + | ++ | none |
| | | hmc (non-neuronal) | ++ | - | - | - | - | - | +++ | none |
| | | muscle | +++ | ++ | ++ | - | - | - | +++ | none |
| Additional neurons in male | | VD12 | +++ | +++ | +++ | +++ | +++ | +++ | +++ | none |
| | | EF1-4 | +++ | +++ | +++ | +++ | +++ | +++ | +++ | none |
| | | R2A | + | + | + | - | - | - | - | ACh |
| | | R6A | ++ | ++ | ++ | - | + | - | - | ACh |
| | | R9B | + | + | + | - | - | +/- | - | none |

Additional sites of *unc-46* and *unc-47* expression are shown in **Table 2**. *unc-25* and *snf-11* are expressed exclusively in the cells shown here.

Color coding in column 3 (same as in **Figure 2E**):

Blue: 'conventional' GABA neurons that synthesize and synaptically release GABA.

Grey: non-conventional GABA neurons which acquire and release GABA by presently unknown means.

Purple: GABA uptake neurons that take up GABA via SNF-11 (based on *snf-11* expression and *snf-11*-dependence of GABA staining).

Green: Non-neuronal GABA uptake cells.

+/- does not consistently stain in all animals. If present, staining is weak.

* Expression as strong as in RIB ('++') in larval stages, but intensity decreases in the adult.

lack the GABA vesicular transporter VGAT/SLC32, called UNC-47 in *C. elegans* (*McIntire et al., 1997*); (b) we stained, with anti-GABA antibodies, animals that lack the sole ortholog of the GABA uptake transporter GAT/SLC6A1, called SNF-11 in *C. elegans* (*Jiang et al., 2005*; *Mullen et al., 2006*); and (c) we analyzed the expression pattern of SNF-11/GAT. We find that GABA staining of the ALA and AVF neurons is abolished in either *unc-47* mutants (no GABA release from other neurons) or *snf-11* mutants (no GABA uptake), further corroborating that these neurons do not synthesize their own GABA, but obtain synaptically released GABA by uptake via the plasma membrane transporter SNF-11/GAT (*Figure 1D,E*). In support of this notion, both ALA and AVF express a fosmid-based reporter that we generated for the *snf-11/GAT* locus (*Figure 2C,D*). We therefore termed the AVF and ALA neurons 'GABA uptake neurons'. Within the nervous system, the *snf-11* fosmid-based reporter is also expressed in the previously characterized, GABA-synthesizing RME and AVL

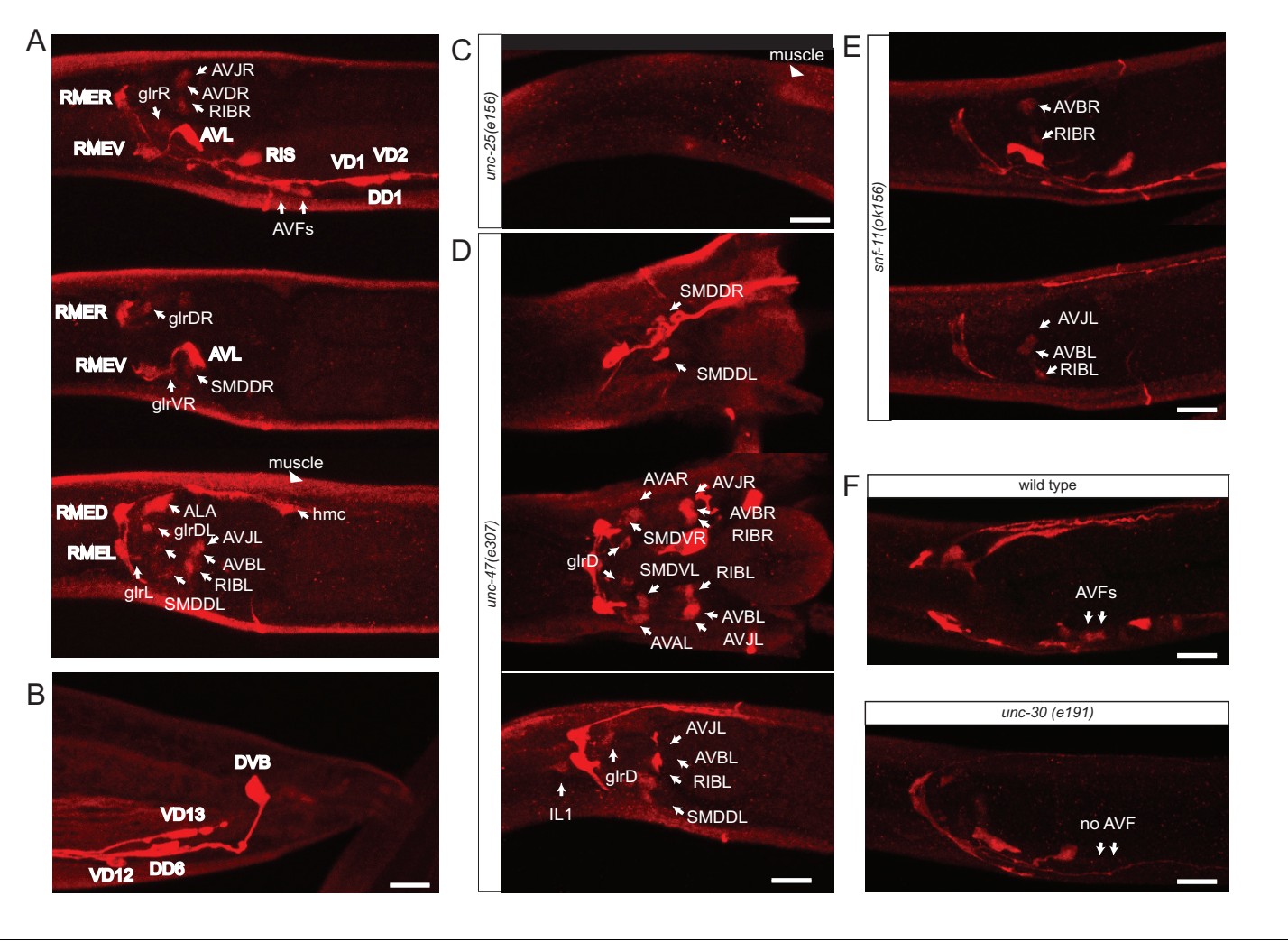

**Figure 1.** Anti-GABA staining defines the GABAergic nervous system of *C. elegans.* (**A**) Three different optical sections of the head of a single adult hermaphrodite, stained with anti-GABA antibodies. Previously identified GABA neurons (***McIntire et al., 1993b***) are shown in bold. (**B**) Anti-GABA staining in the tail of an adult hermaphrodite. (**C**) Anti-GABA staining in the head of an adult *unc-25/GAD(-)* hermaphrodite. Absence of GABA staining illustrates that GABA staining is specific. Staining in muscles is diminished but not eliminated completely. (**D**) Two different optical sections (ventral/ dorsal) and a lateral view of the head of a GABA-stained adult *unc-47/VGAT(-)* hermaphrodite, which illustrates the dependence of some GABA-positive cells on vesicular GABA secretion. Only the newly identified GABA neurons are labeled. SMD is more strongly stained in *unc-47* mutants compared to wild type animals. The hmc, ALA and AVF are not stained. We also observed weak GABA staining in the IL1 neurons of *unc-47(-)* animals, but not in wild-type animals. (**E**) Two different optical sections of the head of a GABA-stained adult *snf-11/GAT(-)* hermaphrodite, which illustrates the dependence of some GABA-positive cells on GABA uptake. Only the newly identified GABA neurons are labeled. hmc, GLR, ALA and AVF are not stained. (**F**) Anti-GABA staining of AVF neurons is absent in *unc-30* mutant animals. (**A–F**) (Scale bar = 10 μm)

neurons, but not the other 'classic' GABA-synthesizing RIS, DVB or D-type neurons nor in any other neuron in the nervous system (***Table 1***, ***2***, ***Figure 2D***).

Where could AVF and ALA receive GABA from? According to the *C. elegans* connectivity map (***White et al., 1986***), the unpaired ALA neuron is not a direct postsynaptic target of any GABA-positive neurons. However, as assessed by the examination of electron micrographs produced by White et al., the processes of ALA are directly adjacent to the newly identified GABAergic SMD neurons (***White et al., 1986***) (S. Cook, pers.comm.). This indicates that ALA may absorb GABA released by the SMD neurons, thereby modulating GABA transmission between SMD and its target neurons. Consistent with this notion, we find that prevention of GABA release in *unc-47/VGAT* mutants results in a considerable increase in GABA staining in the SMDD and SMDV neurons (***Figure 1D***).

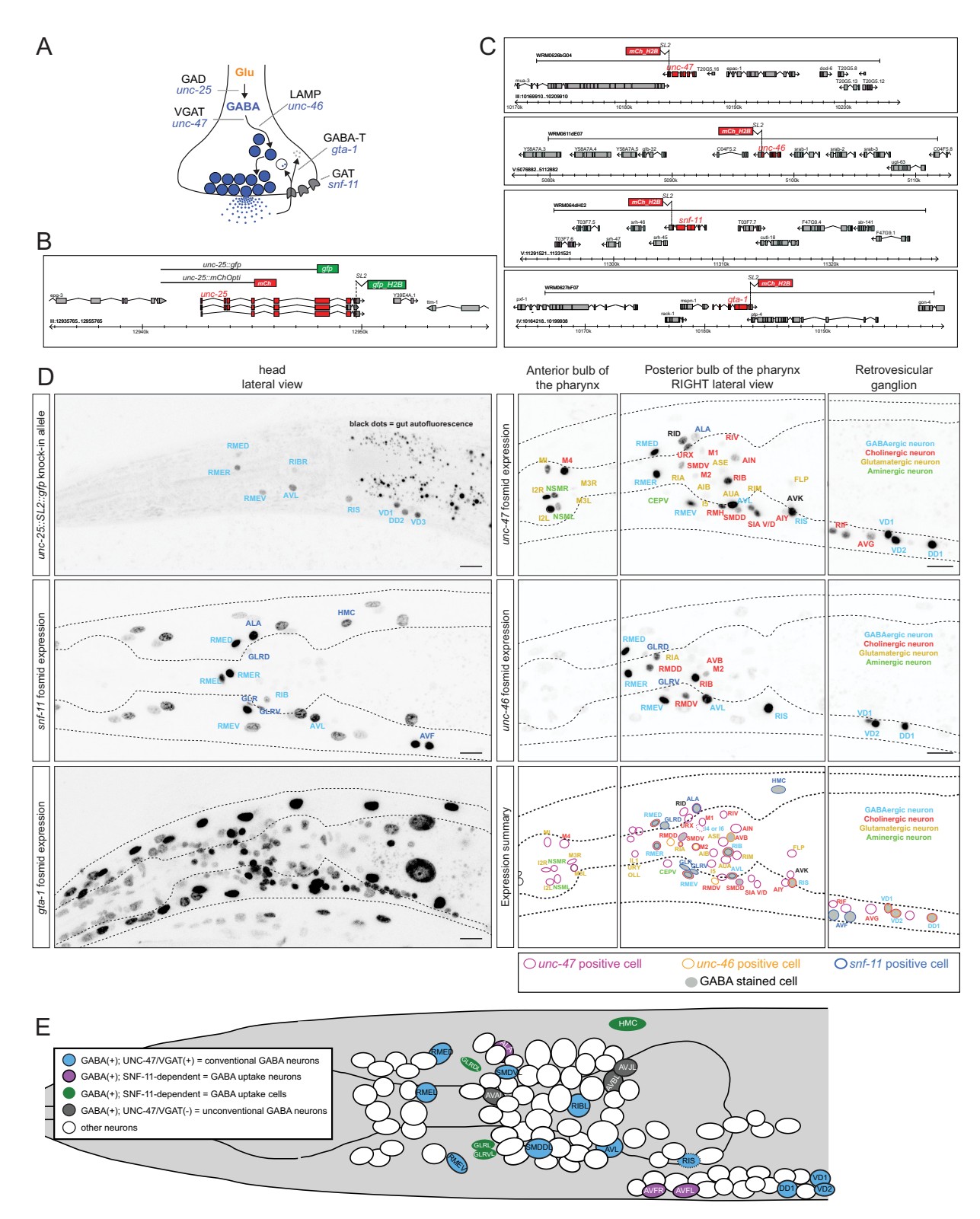

**Figure 2.** Reporter gene constructs recapitulate GABA antibody staining in the nervous system of the *C. elegans* hermaphrodite. (**A**) GABA synthesis, secretion, uptake and degradation pathway. (**B**) *unc-25* reporter transgene schematics and the *unc-25* locus with the knock-in allele *ot867[unc-25::SL2::gfp]* which is targeted to the nucleus to facilitate the detection and the identification of expressing cells. (**C**) Fosmid-based reporter transgenes for *unc-46, unc-47, snf-11* and *gta-1*. Note that the reporters are targeted to the nucleus to facilitate the identification of expressing cells. (**D**) Reporter

*Figure 2 continued on next page*

*Figure 2 continued*

expression. Images are color-inverted. The last panel shows a summary of all reporter expressing neurons. Font color indicates the known neurotransmitter used by the neuron, the colored circle which gene is expressed in the neuron and the grey shade if the neuron is GABA stained. (Scale bar = 10 μm) (E) Summary of all GABA-positive cells in the head of the worm (left view). Color coding same as in *Table 1*. Pharyngeal neurons are not shown. In the remainder of the body of the hermaphrodite, only the D-type motor neurons (some shown here) and the DVB tail neuron are GABA-positive. Dashed circle indicates neurons that present only on the right side of the animals, but shown here on the left side.

The AVF uptake neuron class is located at a different position in the *C. elegans* head ganglia and displays projections very distinct from the ALA neuron. AVF neurons extend processes into the nerve ring, but also into the ventral nerve cord (http://wormatlas.org/neurons/Individual%20Neurons/AVF-frameset.html) where they may take up GABA from proximal GABAergic D-type motor neurons which populate the nerve cord. To test this notion experimentally, we removed GABA selectively from D-type motor neurons, using the *unc-30* mutant strain in which D-type motor neurons fail to produce GABA (*Jin et al., 1994*). In these animals GABA staining in AVF is abolished (*Figure 1F*), suggesting that AVF indeed absorbs GABA from the D-type motor neurons.

Curiously, GABA staining of the AVA, AVB and AVJ neurons, which fail to detectably express *unc-25/GAD::SL2::gfp*, persists in *snf-11/GAT* mutants (consistent with *snf-11* not being expressed in these neurons). Animals lacking the SLC6A transporter most closely related to *snf-11*, the betaine transporter *snf-3* (*Peden et al., 2013*) also still show *unc-25*-dependent GABA staining in AVA, AVB and AVJ (data not shown). These neurons therefore either express very low levels of *unc-25/GAD* that are not detectable via *gfp*-tagging of the *unc-25* locus or these neurons employ non-conventional GABA uptake mechanisms.

## VGAT expression suggests the existence of 'recycling neurons' and 'clearance neurons'

We next sought to assess which of the GABA-positive neurons have the ability to synaptically release GABA via the canonical vesicular GABA transporter VGAT/UNC-47 (*McIntire et al., 1997*). To this end, we generated a fosmid-based *unc-47/VGAT* reporter construct which contains considerably more sequence information than previously described reporter constructs. This reporter shows a much broader pattern of expression compared to the original reporter (*McIntire et al., 1997*); it is expressed in the original set of GABAergic neurons and also in the newly identified GABA-positive ALA, RIB, SMDD and SMDV neurons, suggesting that these neurons not only contain GABA but can also synaptically release it (*Table 1*; *Figure 2D*).

We could not detect *unc-47* fosmid reporter expression in the AVF neurons, which do not synthesize but only take up GABA, indicating that these neurons may only function to clear, but not re-release GABA. We cannot exclude the possibility that GABA may be released by non-conventional mechanisms or by other members of the large SLC transporter family encoded by the *C. elegans* genome (*Hobert, 2013*). We also could not detect *unc-47/VGAT* in the GABA-positive AVA, AVB and AVJ neurons. These neurons may use other members of the solute carrier family for GABA transport into synaptic vesicles or they may use non-conventional GABA release mechanisms, which are also thought to exist in the vertebrate CNS (*Koch and Magnusson, 2009*). For example, the bestrophin ion channel has been shown to mediate GABA release from glia (*Lee et al., 2010*) and there are multiple, uncharacterized bestrophin genes in the *C. elegans* genome (*Hobert, 2013*).

Beyond the above-mentioned GABA-positive cells, the *unc-47* reporter fosmid is expressed in a substantial number of additional neurons (*Figure 2D*, summarized in *Table 2*). Since those neurons are not GABA positive, *unc-47/VGAT* may transport an as yet unknown neurotransmitter in these neurons (perhaps glycine, whose use as a neurotransmitter in *C. elegans* is unresolved; *Hobert et al., 2013*). Alternatively, *unc-47* may not have a vesicular transport function in these neurons, a hypothesis that we base on the expression pattern of a fosmid-based reporter that we generated for the *unc-46* locus. *unc-46* encodes a LAMP-like protein required for the vesicular localization of UNC-47/VGAT (*Schuske et al., 2007*). The *unc-46/LAMP* fosmid-based reporter is expressed in most neurons that are GABA(+) and UNC-47(+) (*Table 1*; *Figure 2D*, *Table 2*) but is not expressed in most of the GABA(−) neurons that express *unc-47/VGAT* (*Table 2*).

**Table 2.** Summary of *unc-47/VGAT* and *unc-46/LAMP*-expressing neurons in GABA-negative neurons. Expression in GABA-positive neurons is shown in **Table 1**. +/-, + and ++ represent relative expression levels. *snf-11/GAT* is expressed exclusively in the cells shown in **Table 1**.

| Cells | anti-GABA | *unc-47* fosmid reporter | *unc-46* fosmid reporter | *snf-11* fosmid reporter | other neuro-transmitter |
|---|---|---|---|---|---|
| AIB | - | +/- | - | - | Glu |
| AIN | - | ++ | - | - | ACh |
| AIY | - | + | - | - | ACh |
| ALN | - | + | - | - | ACh |
| ASE | - | + | - | - | Glu |
| AUA | - | +/- | - | - | Glu |
| AVG | - | + | - | - | ACh |
| AVK | - | + | - | - | peptidergic |
| CEPV | - | + | - | - | dopamine |
| DVA | - | ++ | - | - | ACh |
| FLP | - | + | - | - | Glu |
| IL1 | - | +/- | - | - | Glu |
| OLL | - | +/- | - | - | Glu |
| PLN | - | + | - | - | ACh |
| PVT | - | ++ | - | - | none |
| RIA | - | +/- | ++ | - | Glu |
| RID | - | ++ | - | - | none |
| RIF | - | ++ | - | - | ACh |
| RIM | - | +/- | - | - | Glu |
| RIV | - | + | - | - | ACh |
| RMDD/V | - | - | ++ | - | ACh |
| RMH | - | ++ | - | - | ACh |
| SDQ | - | + | - | - | ACh |
| SIA | - | ++ | - | - | ACh |
| URX | - | ++ | - | - | ACh |
| URY | - | +/- | - | - | Glu |
| MI | - | ++ | - | - | Glu |
| M1 | - | + | - | - | ACh |
| M2 | - | + | + | - | ACh |
| M3 | - | + | - | - | Glu |
| M4 | - | ++ | - | - | ACh |
| I2 | - | ++ | - | - | Glu |
| I4 or I6 | - | ++ | - | - | none |
| I5 | - | +/- | - | - | Glu |
| NSM | - | + | - | - | 5HT |

Lastly, we examined whether the enzyme that degrades GABA, GABA transaminase (GABAT), is expressed and perhaps even enriched in GABA uptake neurons. *C. elegans* contains a single ortholog of GABAT, termed *gta-1*. We find a fosmid-based reporter of *gta-1* to be ubiquitously expressed (*Figure 2D*), which mirrors the very broad tissue distribution of vertebrate GABAT and is consistent with GABAT using substrates other than GABA (*Jeremiah and Povey, 1981*).

In conclusion, we have added another seven GABA-positive neuron classes to the previous list of six GABA-positive neuron types (*Table 1*; *Figure 2E*). One of these neuron classes, the RIB neurons, is a 'conventional' GABA neuron similar to the previously characterized GABA-positive neurons, in that it likely synthesizes and synaptically releases GABA. Two neuron classes (ALA and AVF) are GABA uptake neurons that acquire GABA from neighboring cells to either simply remove GABA (AVF) or possibly also reuse GABA (ALA). Four other neuron classes (SMDD/V, AVA, AVB and AVJ), three of them previously shown to be cholinergic (SMDD/V, AVA, AVB; [*Pereira et al., 2015*]) acquire GABA by as yet unknown means. One of them (SMDD/V) may synaptically release GABA (based on *unc-47/VGAT* expression and increased staining in *unc-47(-)* animals), but whether the other neurons (AVA, AVB, AVJ) employ GABA for neurotransmission is presently not clear.

## GABA in non-neuronal cells

In addition to neuronal staining, we also detected GABA in three classes of non-neuronal cells, the head mesodermal cell (hmc), the glia-like GLR cells and body wall muscle (*Figure 1A*). Expression in all these cells depends on *unc-25/*GAD (*Figure 1C*; note that residual staining remains in body wall muscle, perhaps an indication of alternative GABA-synthesis pathways; [*Kim et al., 2015*]). The body wall muscle may take up GABA after release by the D-type motor neurons which innervate body wall muscle. Consistent with this notion, *snf-11/GAT* is expressed in body wall muscle (*Figure 2D*) (*Mullen et al., 2006*).

The hmc is an intriguing cell with no previously ascribed function. It is located above the posterior bulb of the pharynx and extends processes along the ventral and dorsal nerve cord (http://www.wormatlas.org/ver1/handbook/mesodermal.htm/hmc.htm). The processes of the hmc are in proximity to the processes of a number of GABAergic neurons (RMED/V and D-type motor neurons) (*Hall and Altun, 2007*) and therefore in the proper place to clear GABA. We indeed find that hmc expresses the GABA uptake transporter *snf-11/GAT* (*Figure 2D*) and that GABA staining of the hmc is abolished in *snf-11/GAT* mutants (*Figure 1E*). We also find that GABA staining in the hmc is reduced in *unc-30* mutants (in which D-type motor neurons fail to be specified; [*Jin et al., 1994*]), indicating that the sources of GABA in the hmc are indeed the D-type motor neurons. Since we do not detect *unc-47/VGAT* expression in the hmc, the hmc likely operates as a GABA clearance cell, like the AVF neurons.

The glia-like GLR cells are another intriguing GABA-positive cell type. GLR cells, which have no assigned function yet, are located directly adjacent to the nerve ring (http://www.wormatlas.org/ver1/handbook/mesodermal.htm/glr.htm). Each GLR cell extends a thin, sheet-like process that lies inside the nerve ring. Like the AVF and ALA neurons and the non-neuronal hmc, the GLR cells express *snf-11/GAT* (*Figure 2*) and GABA-staining is strongly reduced in *snf-11/GAT* mutants (*Figure 1*). Curiously, staining is still observed in *unc-47/VGAT* mutants. In these mutants, GABA may accumulate in RME neurons which are heavily gap-junctioned with the GLR glia cells (*White et al., 1986*) and GABA may pass through these gap junctions. The passive transfer of neurotransmitters through gap junctions has been termed 'neurotransmitter coupling' and occurs, for example, in the amacrine and bipolar cells of the vertebrate retina (*Vaney et al., 1998*).

## GABAergic neurons in the male nervous system

In males, we observed the same set of GABA-positive neurons as observed in hermaphrodites. In addition, GABA staining is observed in one prominent class of male-specific interneurons , the EF neurons, which had previously no neurotransmitter assigned (*Figure 3B*). The EF1 and EF2 interneurons are located in the dorsorectal ganglion, whereas EF3 and the rarely generated EF4 are in the preanal ganglion (*Figure 3B*). The EF neurons are so-called 'type II interneurons', which relay sensory information from male-specific tail sensory structures into the sex-shared nervous system in the head (*Jarrell et al., 2012*). All the EF neurons also express *unc-25/GAD*, *unc-47/VGAT* and *snf-11/*GAT, demonstrating that these neurons synthesize, release and reuptake GABA (*Figure 3D*). Apart from the EF neurons, three additional male-specific sensory neurons stain with anti-GABA antibodies, the R2A, R6A and R9B pairs of ray neurons (*Figure 3B*), but like the head AVA, AVB and AVJ, these neurons do not express *unc-25/GAD* and only R6A expresses *unc-47/VGAT*. Expression in all male tail neurons is unaffected in *snf-11/GAT* mutants (data not shown).

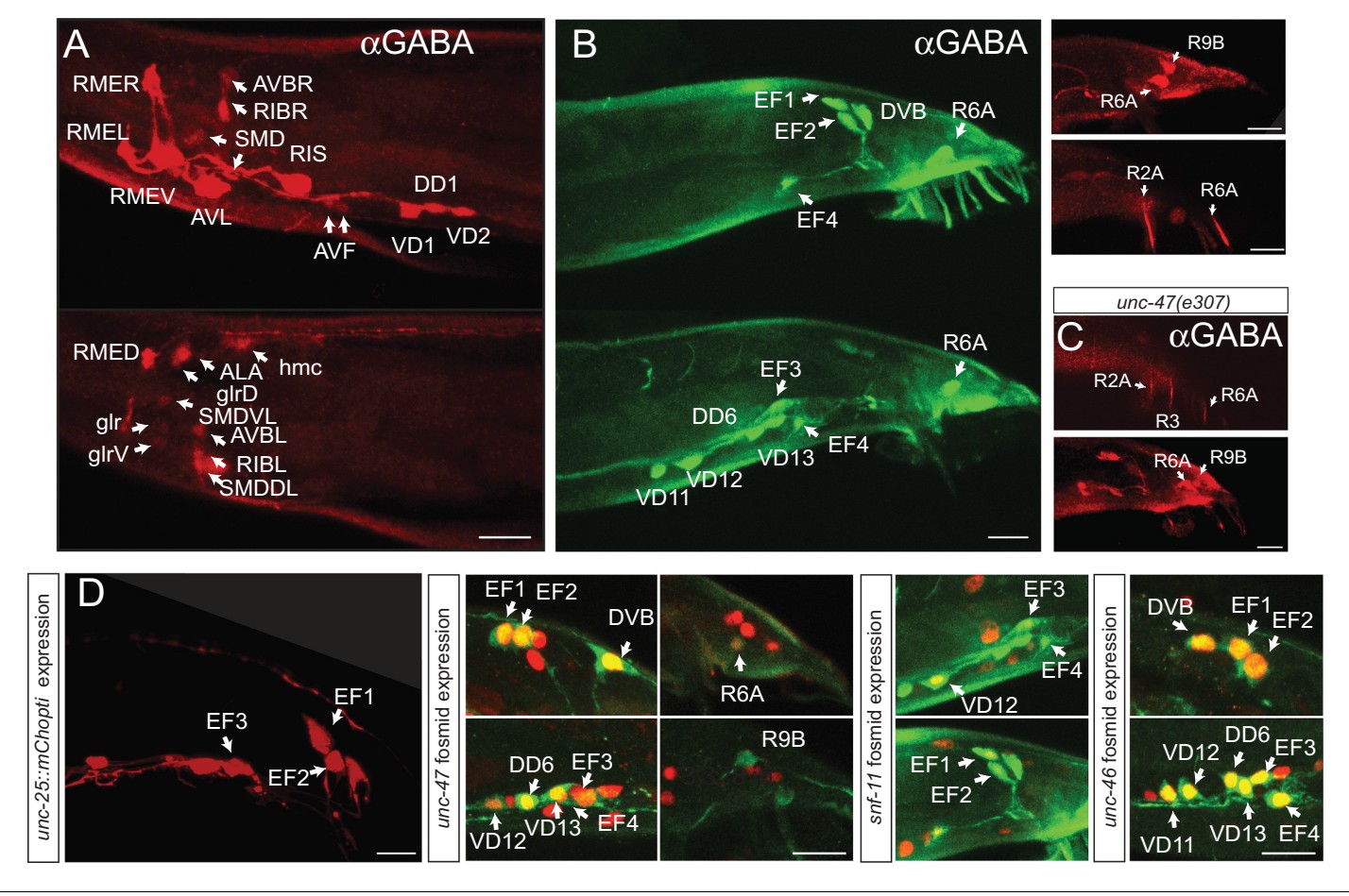

**Figure 3.** GABAergic neurons in the male. (A–C) Anti-GABA staining of adult males in the head and tail in different optical sections. (Scale bar = 10 μm) (**A**) In adult male head, same neurons as in the hermaphrodite head are GABA-positive. (**B**) Seven male-specific neurons are GABA-stained: EF1-4, R2A, R6A and R9B (white arrows; two optical section with green fluorescent secondary antibody in left two panels with ray neuron staining being background and red fluorescent secondary antibody in upper right panels with ray neuron showing no background staining). (**C**) In *unc-47* mutants, GABA staining in R2A, R6A and R9B persists, indicating that these neurons do not acquire GABA via secretion from other neurons. In addition, we observe GABA staining in an extra ray neuron, R3, that we cannot detect in wild-type animals. (**D**) Reporter gene expression in adult male tail. Left panel: *unc-25* reporter, other panel shows red fosmid reporter (nucleus) and green anti-GABA staining. Reporter constructs as shown in *Figure 2*. EF1-4 expressed *unc-25/GAD, unc-47/VGAT* and *snf-11/GAT. unc-47/VGAT* is expressed in R6A but not in R2A or R9B. White arrow indicates GABA stained cells colocalizing with the reporter. *snf-11/GAT* is present in the male VD12 but not in the hermaphrodite. *unc-25/GAD* transcriptional reporter shows the same expression pattern as the *gfp* knock-in allele in the male tail (Scale bar = 10 μm).

Lastly, we noted an intriguing sexual dimorphism of *snf-11/GAT* expression in the VD12 neuron, which is generated in both sexes and produces GABA in both sexes, based on *unc-25/GAD* expression and GABA staining. In hermaphrodites, none of the 13 VD neurons express *snf-11/GAT*, but in males, VD12 expresses *snf-11/GAT* (*Figure 3*). VD12 receives a multitude of male-specific synaptic inputs (*Jarrell et al., 2012*), but receives no synaptic input from GABA-positive male tail neurons. Perhaps it is particularly relevant to limit the amount of GABA released by VD12 onto muscle in the male, but not hermaphrodite tail.

## Synaptic connectivity of GABA-positive neurons

We used the synaptic wiring diagram of the hermaphrodite elucidated by John White and colleagues (*Varshney et al., 2011*; *White et al., 1986*) (www.wormwiring.org) to assess the extent of GABAergic synaptic innervation throughout the entire nervous system. Of the 118 anatomically defined neuron classes of the hermaphrodite, 47 neuron classes are innervated by GABA(+); UNC-47(+) neurons

**Table 3.** Synaptic targets of GABA(+); UNC-47(+) neurons. Coloured box indicates that GABAergic output neuron synapses onto this target cell (<5 synapses [grey]; 5-20 synapses [orange]; >20 synapses [pink]) and genes in the boxes represent expression of ionotropic GABA$_A$-type receptor subunits in the target cell (*Figure 4*). For a complete list of GABA receptor (+) neurons, see *Table 4*. Blue shading indicates that the target neurons is GABA(+). Wiring data is from www.wormwiring.org and includes all synapses observed. * = RMDL/R only.

| Targets | | ALA | RIB | RIS | AVL | DVB | RME | SMD | DD/VD |
|---|---|---|---|---|---|---|---|---|---|
| | | **GABAergic output neurons** | | | | | | | |
| muscle | | | | | exp-1, lgc-37, gab-1 [orange] | exp-1, lgc-37, gab-1 [orange] | unc-49, lgc-37, gab-1 [pink] | unc-49, lgc-37, gab-1 [pink] | unc-49, lgc-37, gab-1 [pink] |
| sensory neuron (9 classes) | ASG | [grey] | | | | | | [grey] | |
| | ASJ | | | | | | | | |
| | BAG | | [orange] | | | | | | |
| | CEPD | | | [orange] | | | | | |
| | CEPV | | [orange] | | | | | [grey] | |
| | IL1 | | | | | | [orange] | | |
| | IL2 | | | | | | | lgc-38 | |
| | PHC | | | | | [grey] | | | |
| | OLL | | lgc-38 [grey] | lgc-38 [orange] | | | lgc-38 [grey] | | |
| | URY | | | [orange] | | | | | |
| inter-neuron (22 classes) | AIB | | lgc-38 [orange] | | | | | lgc-38 [grey] | |
| | AIN | | lgc-38 [grey] | | | | | | |
| | AIZ | | lgc-37, lgc-38, gab-1 [grey] | | | | | lgc-37, lgc-38, gab-1 [grey] | |
| | ALN | | | | | | | [grey] | |
| | AUA | | [orange] | | | | | | |
| | AVA | | lgc-37, gab-1 [orange] | | | | | lgc-37, gab-1 [grey] | |
| | AVB | | [grey] | | | | | | |
| | AVD | lgc-35 [grey] | lgc-35 [grey] | | | | | | |
| | AVE | gab-1 [orange] | gab-1 [pink] | gab-1 [orange] | gab-1 [grey] | | | gab-1 [grey] | |
| | AVK | | | [orange] | | | | | |
| | DVC | | [grey] | | | [grey] | | | |
| | PVP | [grey] | | | | [grey] | | | |
| | PVR | | | | | | | lgc-37, gab-1 | |
| | PVW | | | | | lgc-37, gab-1 [grey] | | | |
| | RIA | | [orange] | | | | | [pink] | |
| | RIB | | | [pink] | | | | [grey] | |
| | RIG | | [orange] | | | | | | |
| | RIH | | [grey] | | | | [grey] | | |
| | RIP | | | | | | | | |
| | RIS | | | | | | | | |
| | SAA | | | | | | | [grey] | |

*Table 3 continued on next page*

*Table 3 continued*

| Targets | | GABAergic output neurons | | | | | | | |
|---|---|---|---|---|---|---|---|---|---|
| | | ALA | RIB | RIS | AVL | DVB | RME | SMD | DD/VD |
| Motor neuron (16 classes) | AVL | | | | | | | | |
| | DA | | | | | lgc-37, gab-1, lgc-35 | | | |
| | DD | | | | | | | | |
| | DVB | | | | | | | | |
| | HSN | | | | lgc-37, gab-1 | | | | |
| | PDA | | | | | exp-1, lgc-35 | | | |
| | RIM | | | | | | | | |
| | RIV | | | | | | | lgc-38 | |
| | RME | | | | | | | | |
| | RMD* | gab-1, lgc-37 | gab-1, lgc-37 | gab-1, lgc-37 | | | gab-1, lgc-37 | gab-1, lgc-37 | |
| | RMH | | | | | | | | |
| | SAB | | | | exp-1, lgc-37, gab-1, lgc-36 | | | | |
| | SIA | | | | | | | lgc-37, gab-1 | |
| | SIB | | | | | | | | |
| | SMD | | lgc-37, lgc-38, gab-1 | lgc-37, lgc-38, gab-1 | | | lgc-37, lgc-38, gab-1 | lgc-37, lgc-38, gab-1 | |
| | VC | | | | lgc-37, gab-1 | | | | |
| | VD | | | | | | | | |

(*Table 3*). The RIB and SMD neurons are the GABAergic neurons with the most synaptic outputs (*Table 3*). Both neurons also employ ACh as a neurotransmitter, even though in each case, one neurotransmitter system appears to predominate: The RIB neurons express barely detectable levels of VAChT/ChAT (*Pereira et al., 2015*), while their GABA staining is easily detectable. Conversely, the SMD neurons strongly express VAChT/ChAT (*Pereira et al., 2015*), but their GABA content is comparatively weak and variable.

Given the relatively small number of GABAergic neurons, it is notable that 23 out of the 47 neurons that receive synaptic input from GABAergic neurons receive such inputs from more than one GABAergic neuron; in many cases inputs are received from more than half of all the distinct GABAergic neuron classes (*Table 3*). For example, the RMD head motor neurons are innervated by the GABAergic RME and SMD motor neurons, by the RIS and RIB interneurons and by the ALA sensory neuron. The GABAergic RIS and RIB interneurons co-innervate a number of distinct neuron classes. Apart from the RMD head motor neurons, the OLL and URY head sensory neurons, the AVE command interneuron and the RIM, RME and SMD head motor neurons are co-innervated by RIS and RIB.

Moreover, all except one of the GABA-positive neurons (the unusual ALA GABA uptake sensory neuron), are postsynaptic to other GABAergic neurons (*Table 3*). This observation suggests that so-called 'GABAergic disinhibition' (=GABA-mediated inhibition of an inhibitory GABA neuron), a common organizational principle of inhibitory circuits in the vertebrate CNS (*Roberts, 1974*), may also be broadly occurring in the *C. elegans* nervous system. This hypothesis will need to be experimentally validated.

## Expression of ionotropic GABA receptors

The 47 neurons that are postsynaptic to GABA(+); UNC-47(+) neurons are candidates to express at least one of the seven GABA$_A$-type receptor encoded by the *C. elegans* genome (*Hobert, 2013*). The expression of three of these receptors have been previously examined, *unc-49*, *exp-1* and *lgc-35*

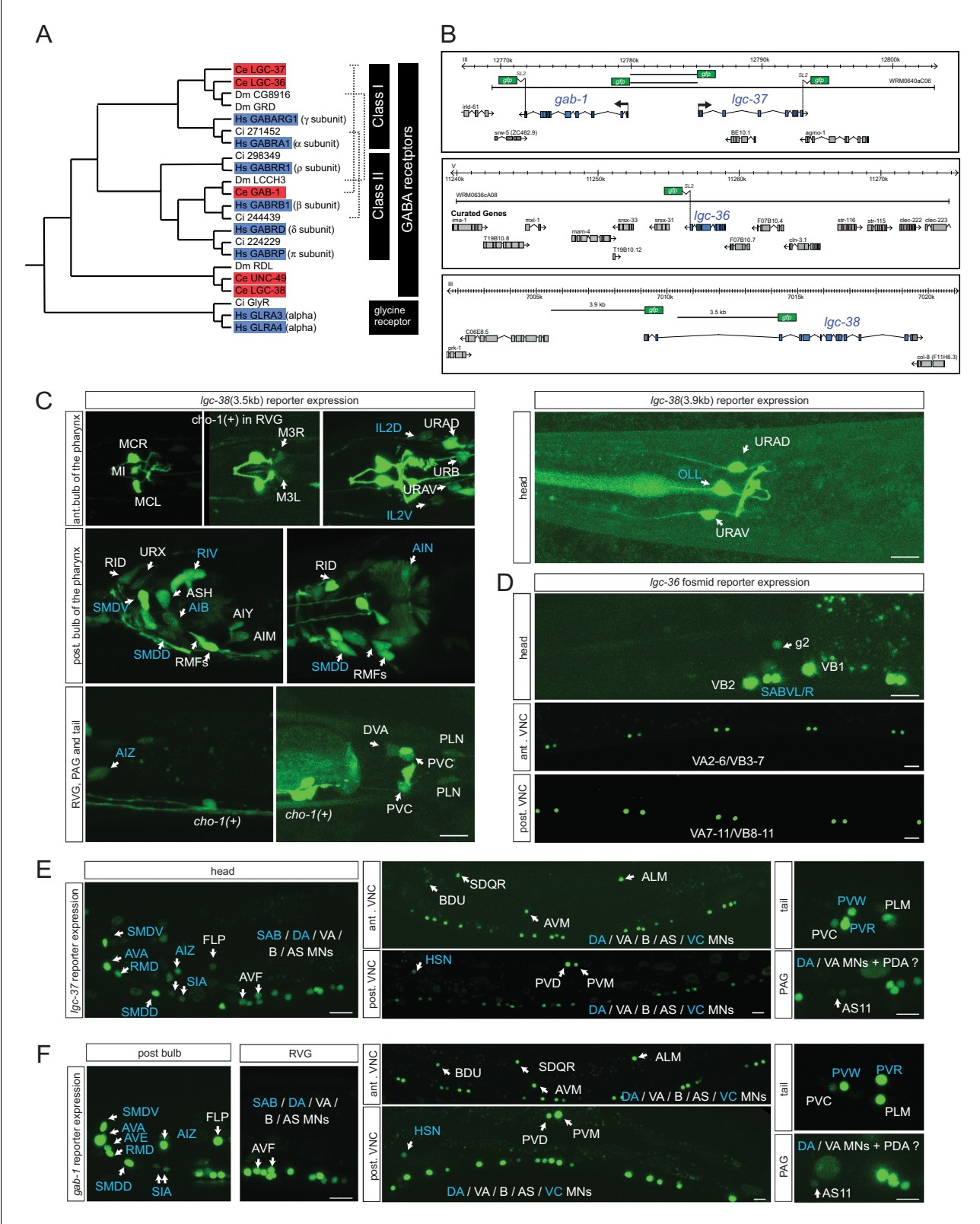

**Figure 4.** Ionotropic GABA receptor expression. (**A**) Phylogeny of the *C. elegans* GABA<sub>A</sub> receptor-encoding genes (red boxes) that are most similar to canonical human GABA<sub>A</sub> receptor-encoding genes (blue boxes). The dendrogram has been adapted from (*Tsang et al., 2007*). Two other bona-fide GABA receptors, EXP-1 and LGC-35, are more distantly related, falling outside this part of the tree. (**B**) GABA reporter transgene schematics. All the fosmids are targeted to the nucleus to facilitate the detection and the identification of expressing cells. (**C–F**) Expression of *lgc-38* (C), *lgc-36* (D), *lgc-37*

*Figure 4 continued*

(E) and *gab-1* (F) reporter constructs. Blue labels indicate neurons innervated by a GABA-positive neuron (see *Table 3*). Cells were identified with *unc-47*, *eat-4* or *cho-1* reporters as landmarks in the background. The transcriptional reporter gene fusions for *lgc-38* (see Materials and methods) do not capture the entire locus and hence they may lack regulatory information but there is no fosmid available that encompasses the entire locus. (Scale bar = 10 μm, VNC: ventral nerve cord, PAG: pre-anal ganglion, RVG: retro-vesicular ganglion; MN = motor neuron)

(*Bamber et al., 1999*; *Beg and Jorgensen, 2003*; *Jobson et al., 2015*) (*Table 4*). However, the expression of the most canonical GABA$_A$-type receptors in the worm genome (shown in *Figure 4A*) (*Tsang et al., 2007*), the two alpha-subunit type GABA$_A$ receptors LGC-36 and LGC-37 and the beta-subunit GABA$_A$ receptor GAB-1, has not previously been reported. The expression of a UNC-49-related GABA receptor, *lgc-38*, has also not yet been examined. We examined the expression of all of these four putative GABA receptor-encoding genes, using reporter gene fusions. Of particular note is the head-to-head orientation of the alpha-subunit-encoding *lgc-37* gene and the beta sub-unit-encoding *gab-1* gene (*Figure 4B*); strikingly, this genomic organization is conserved from nematodes to vertebrates (*Darlison et al., 2005*; *Tsang et al., 2007*). This organization suggests that both genes employ the same *cis*-regulatory elements to ensure co-expression.

All four reporter transgenes are expressed in a restricted number of head and tail neurons (*Figure 4B–F*; summarized in *Table 4*). *lgc-36* shows the most restricted expression (only in SABVL/R head motor neurons and a subset of ventral nerve cord motor neurons, the VA and VB neurons). As expected by their above mentioned genomic organization, *lgc-37* and *gab-1* fosmid reporters are indeed co-expressed in 25 neuron classes (*Table 4*). The only neuron where just one of the two fosmid reporters is expressed in, *gab-1*, is the AVE command interneuron (*Figure 4F*). All the regulatory elements for this co-expression appear to be harbored by the shared 5' intergenic region, since transcriptional reporter fusions that contain only the 5' intergenic region in either orientation (shown in *Figure 4B*) show an expression pattern similar to the fosmid reporter, with the sole exception of the AVE neuron, the only neuron that expresses the *gab-1* fosmid reporter, but not the *lgc-37* fosmid reporter (data not shown). Regulatory elements for AVE expression must therefore resides elsewhere in the *gab-1* locus.

The co-expression of *gab-1* and *lgc-37*, as well as other patterns of GABA receptor co-expression (summarized in *Table 4*) suggest that the respective receptors form heteromeric receptors in the respective neuron types. We mapped neurons that express any of the GABA$_A$-type receptors onto a matrix of neurons innervated by GABA neurons, as shown in *Table 3* (and as indicated by blue letter coding in *Figure 4C–F*). This expression data provides a possible starting point to disrupt GABAergic signaling in a synapse-specific manner.

In addition to being expressed in neurons that show anatomic innervation by GABAergic neurons, it is clearly evident that GABA$_A$-type receptors are also expressed in neurons that are not anatomically connected to GABA-positive neurons (*Figure 4B–D*, white labels; all neurons listed in *Table 4*). These receptors may simply not function as GABA receptors or, more interestingly, these receptors may mediate GABA spillover transmission, a phenomenon observed in the vertebrate CNS as well as in *C. elegans* (*Jobson et al., 2015*; *Rossi and Hamann, 1998*). Specifically, the previously described expression patterns of *exp-1* and *lgc-35* reveal expression in non-GABA-innervated neurons (*exp-1*: ADE, RID; *lgc-35*: A/B-type motor neurons, DVA, PVT, AIY). Our expression analysis further extends this notion (*Figure 4C–F*), identifying expression of, for example, several of the GABA receptor-encoding genes in the cholinergic A/B-type neurons that were previously reported to receive spillover GABA signals (*Jobson et al., 2015*). Spillover transmission may even extend into the pharyngeal nervous system, where we detect GABA$_A$-type receptor expression (*Table 4*), but no GABA staining.

## The Tailless/Tlx orphan nuclear receptor NHR-67 controls the GABAergic identity of a diverse set of GABAergic neurons

We used the expanded map of the GABAergic nervous system as a starting point to elucidate transcriptional regulatory programs that specify GABAergic neuron identity. Previously, the effect of a number of transcription factors on GABAergic identity has been described, but in most cases, the analysis has either been limited to a few markers or a number of important questions about the

**Table 4.** Summary of expression of GABA$_A$-type receptors. *gab-1*, *lgc-37*, *lgc-38* and *lgc-36* expression is from this paper (**Figure 4**), *exp-1* is from **Beg and Jorgensen (2003)**, *lgc-35* from **Jobson et al. (2015)** and *unc-49* from **Bamber et al. (1999)**. Cells that receive GABAergic synaptic input (as shown in wormwiring.org) are boxed blue. Others likely receive GABA spillover signals. For *lgc-38*, all expressing cells shown are observed with the 3.5 kb reporter fusion, except for OLL, which only expresses the 3.9 kb fusion; URA expresses both.

| cell type | gab-1 | lgc-37 | lgc-38 | lgc-36 | exp-1 | lgc-35 | unc-49 |
|---|---|---|---|---|---|---|---|
| ADE *(boxed)* | | | | | ▓ | | |
| AIB *(boxed)* | | | ▓ | | | | |
| AIM | | | ▓ | | | | |
| AIN *(boxed)* | | | ▓ | | | | |
| AIY | | | ▓ | | | ▓ | |
| AIZ *(boxed)* | ▓ | ▓ | ▓ | | | | |
| ALM | ▓ | ▓ | | | | | |
| ASH | | | ▓ | | | | |
| AVA *(boxed)* | ▓ | ▓ | ▓ | | | | |
| AVD *(boxed)* | | | | | | ▓ | |
| AVE *(boxed)* | ▓ | ▓ | | | | | |
| AVF | ▓ | ▓ | | | | | |
| AVM | ▓ | ▓ | | | | | |
| AS | ▓ | | | | | | |
| BDU | ▓ | | | | | | |
| DA *(boxed)* | ▓ | ▓ | | | | ▓ | |
| DB | ▓ | ▓ | | | | ▓ | |
| DVA | | | ▓ | | | | |
| FLP | ▓ | | | | | | |
| HSN *(boxed)* | ▓ | ▓ | | | | | |
| IL2 *(boxed)* | | | ▓ | | | | |
| OLL *(boxed)* | | | ▓ | | | | |
| PDA *(boxed)* | | | | | ▓ | ▓ | |
| PLM | ▓ | | | | | | |
| PLN | | | ▓ | | | | |
| PVC | ▓ | ▓ | ▓ | | | | |
| PVD | ▓ | | | | | | |
| PVM | ▓ | | | | | | |
| PVR *(boxed)* | ▓ | | | | | | |
| PVT | | | | | | ▓ | |
| PVW *(boxed)* | ▓ | ▓ | ▓ | | | | |
| RID | | | ▓ | | | ▓ | |
| RIV *(boxed)* | ▓ | | | | | | |
| RMD* *(boxed)* | ▓ | ▓ | ▓ | | | | |
| RMF | | | ▓ | | | | |
| SABD *(boxed)* | ▓ | | | | | ▓ | |
| SABV *(boxed)* | ▓ | | | ▓ | | | |
| SDQR † | ▓ | | | | | | |
| SIA *(boxed)* | ▓ | | | | | | |
| SMD *(boxed)* | ▓ | ▓ | ▓ | | | | |

*Table 4 continued on next page*

*Table 4 continued*

| cell type | gab-1 | lgc-37 | lgc-38 | lgc-36 | exp-1 | lgc-35 | unc-49 |
|---|---|---|---|---|---|---|---|
| URA | | | | | | | |
| URB | | | | | | | |
| URX | | | | | | | |
| VA | | | | | | | |
| VB | | | | | | | |
| VC | | | | | | | |
| M3 | | | | | | | |
| MC | | | | | | | |
| MI | | | | | | | |
| muscle | | | | | | | |

* Expression is only observed in RMDL/R, not RMDD/V.
† Asymmetrically expressed in only SDQR, not SDQL.

specificity of the involved factors has remained unanswered. We describe the functions of these factors systematically in this and the ensuing sections.

Sarin *et al.* have previously shown that the Tailless/Tlx orphan nuclear receptor *nhr-67* is expressed in the GABAergic RMEs, RIS and AVL neurons and that *nhr-67* expression is maintained in these neurons throughout adulthood in the RME and RIS neurons (*Sarin et al., 2009*). Consistent with a possible role of *nhr-67* as a regulator of GABAergic identity, it was also reported that loss of *nhr-67* affected expression of *unc-47/VGAT* in RME, AVL and RIS (*Sarin et al., 2009*). However, whether *nhr-67* affects GABA synthesis, or the expression of other GABAergic identity features was not tested. We first isolated an unambiguous molecular null allele of *nhr-67*, *ot795*, using *Mos* transposon-mediated gene deletion (MosDEL; see Experimental Procedures) (*Figure 5A*). *nhr-67(ot795)* animals display an embryonic/L1 arrest phenotype. We stained these null mutant animals for GABA and examined *unc-25/GAD*, *unc-47/VGAT* and *unc-46/LAMP* expression in these mutants. We found abnormalities in the expression of all markers in all three, normally *nhr-67*-expressing neuron classes, AVL, RME and RIS (*Figure 5B*). In the AVL neuron, defects in *nhr-67* mutants are modest, but as we will describe in the next section, these defects are strongly enhanced by removal of a factor that likely cooperates with *nhr-67*.

## *nhr-67* cooperates with distinct homeobox genes in distinct GABAergic neurons

Two previous studies had implicated two homeobox genes in the regulation of GABA identity of a subset of the *nhr-67* expressing neurons, the Prd-type homeobox gene *ceh-10* and the LIM homeobox gene *lim-6*. *ceh-10* was found to affect GABA staining and *unc-25/GAD* expression in the RMED neurons (where *ceh-10* was reported to be expressed) (*Forrester et al., 1998*; *Huang et al., 2004*). We extended this previous finding by demonstrating that in *ceh-10* mutants, not only the *unc-25/GAD* reporter, but also the *unc-47/VGAT*, *unc-46/LAMP* and *snf-11/GAT* reporters fail to be expressed in RMED (*Figure 5E*). Absence of expression of these reporters is not reflective of lineage defects, since we find that in *ceh-10* hypomorphic mutants in which GABA staining is also severely affected, the RMED neurons are nevertheless formed, as assessed by the intact expression of a pan-neuronal marker (*Figure 5F*). In addition to affecting expression of GABAergic markers, *ceh-10* also affects the expression of an *nhr-67* fosmid based reporter in RMED (*Figure 5—figure supplement 1*).

The LIM homeobox gene *lim-6* was found to control *unc-25* expression in the RIS, AVL and DVB neurons (*Hobert et al., 1999*). We corroborated the impact of *lim-6* on GABA identity by anti-GABA staining of *lim-6* null mutant animals (which had not previously been done), finding that GABA staining is affected in AVL, RIS and DVB (*Figure 5C*). Since *nhr-67* null mutants did not display fully penetrant defects in the AVL and RIS neurons, we tested whether *nhr-67* and *lim-6* may collaborate in the

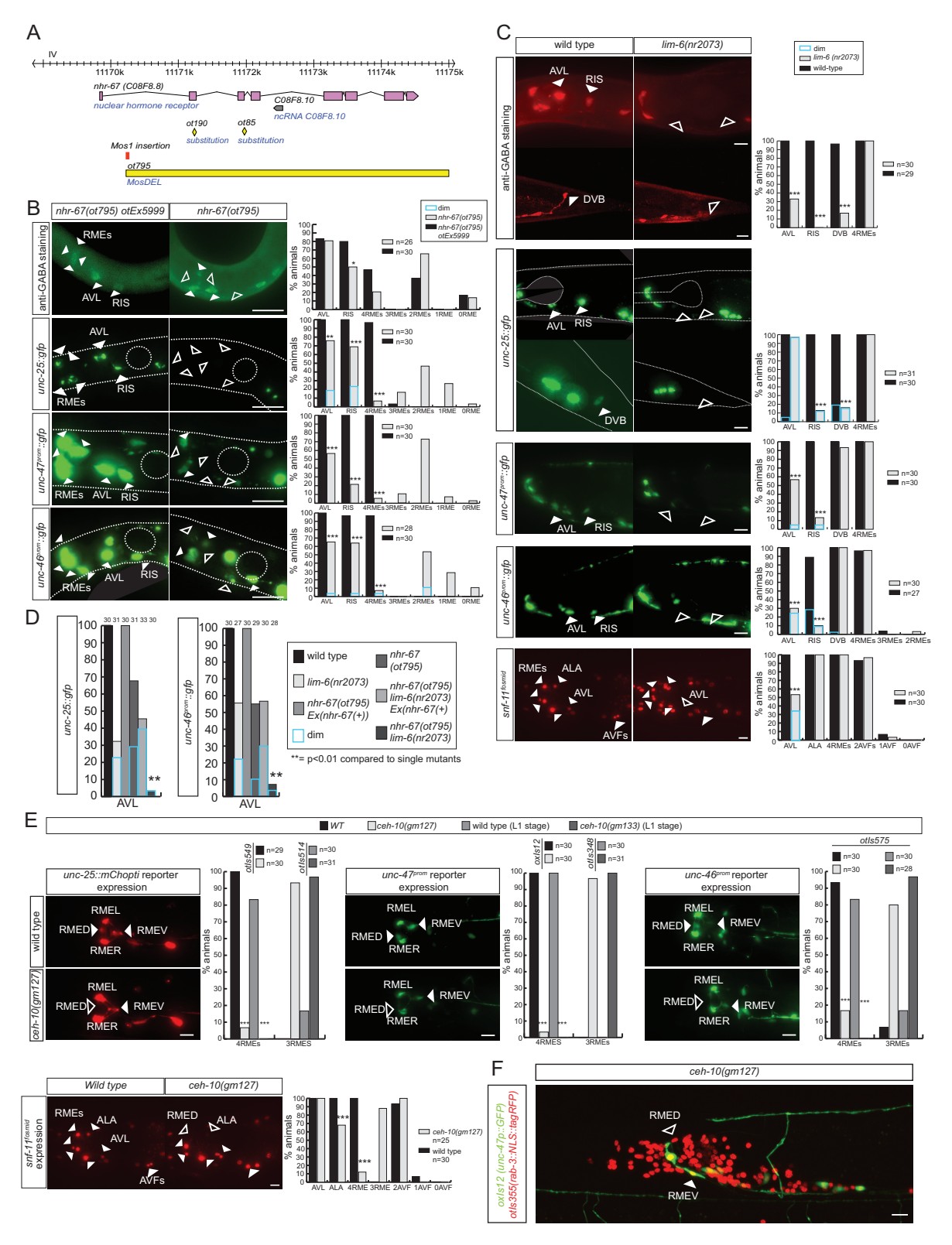

**Figure 5.** *nhr-67* cooperates with distinct homeodomain proteins in distinct GABAergic neuron types. (A) *nhr-67* locus with newly generated null allele. (B–C) Anti-GABA staining and expression of GABA pathway genes in *nhr-67* and *lim-6* null mutant backgrounds (Fisher exact test for all the data except for GABA staining in *nhr-67* mutant where Chi-square test was used; *p<0.05, **p<0.01 and ***p<0.001 compared to single mutants; for expression in RMEs statistics were performed only on the category 4RMEs)) (B) AVL, RIS and RMEs are affected in *nhr-67* null mutants. *nhr-67(ot795)* animals are not

*Figure 5 continued on next page*

*Figure 5 continued*

viable and *otEx5999* is an extrachromosomal array that contains copies of the wild-type *nhr-67* locus as well as an array marker (*unc-47^{prom}::mChOpti*). Animals that either contain or do not contain this array were scored for mutant phenotypes at the L1 stage. (**C**) *lim-6* null mutant animals display defects in GABAergic neuron identity of AVL, RIS and DVB. (**D**) *nhr-67* and *lim-6* interact genetically in the AVL neuron. Expression of *unc-25/GAD* and *unc-46/LAMP* in AVL were scored at L1 in the different mutant background. (Chi-square test, **p<0.01 compared to single mutants) (**E**) *ceh-10* mutant animals display defects in RMED GABAergic neuron identity. Expression of GABA pathway genes in null mutants (*gm133*) was scored as L1, when they arrest development. Viable *gm127* hypomorphic mutants were scored as adults. Wild type data in panel C are re-iterated in panel E. (**F**) *ceh-10* does not affect the generation of RMED as assessed by expression of a pan-neuronal marker in red but affects GABAergic fate as assessed by the absence of the *unc-47/VGAT* reporter in green. (**A–F**) (plain arrow head indicates a neuron presence and empty arrow head its absence) (Scale bar = 10 μm)

The following figure supplement is available for figure 5:

**Figure supplement 1.** Regulatory relation of *nhr-67* and other homeobox genes.

specification of these neurons. We find that in *nhr-67; lim-6* double null mutants, the AVL neurons show synergistic defects in the GABAergic identity specification (**Figure 5D**).

*lim-6* expression is not affected in AVL and RIS of *nhr-67* mutants, but *lim-6* (as well as *ceh*-10) controls *nhr-67* expression (**Figure 5—figure supplement 1**). Taken together, *nhr-67* appears to collaborate with distinct homeobox genes in distinct neurons and appears to be regulated by these factors. Since both *ceh-10* and *lim-6* remain expressed throughout the life of the respective neurons, we surmise that *ceh-10* and *lim-6* induce a critical cofactor (*nhr-67*) that they then work together with. In other cases of collaborating transcription factors, it has also been demonstrated that one factor acts upstream of the other to then cooperate with the induced factor (e.g. *ttx-3* induces *ceh-10* expression and both factors then cooperate to drive cholinergic identity of the AIY neurons; [**Bertrand and Hobert, 2009**; **Wenick and Hobert, 2004**]; or *unc-86* induces *mec-3* expression to then cooperate in the specification of touch receptor neurons [**Duggan et al., 1998**]).

## The homeobox gene *tab-1* controls GABAergic identity of the RMEL/R neurons

While the *lim-6* and *ceh-10* homeobox genes may work together with *nhr-67* in subsets of GABAergic neurons to control GABAergic identity, no cooperating factor for *nhr-67* function in the RMEL/R neurons was apparent. We screened several dozen homeobox mutants for defects in GABA staining without success (data not shown) and then screened for EMS-induced mutants in which *unc-47* expression in the RME neurons was absent (see Materials and methods). We identified a mutant allele, *ot796,* in which *unc-47* failed to be expressed in the left and right RME neurons. Whole genome sequencing revealed that *ot796* contains a splice site mutation in the *tab-1* locus (**Figure 6A**; **Figure 6—figure supplement 1**; **Table 5**). The *ot796* mutant allele fails to complement the GABA differentiation defects of other mutant alleles of *tab-1* and three additional alleles of *tab-1* display similar *unc-47* expression defects as *ot796* (**Table 5**). *tab-1* mutants also fail to properly express *unc-25* and *unc-46* in RMEL/R and fail to antibody-stain for GABA (**Figure 6B**). A *tab-1* fosmid-based reporter is expressed in both the left and right RME neurons (**Figure 6C**).

*tab-1* encodes the sole *C. elegans* ortholog of the *Drosophila* Bsh homeobox gene and the vertebrate Bsx genes (**Pang and Martindale, 2008**). *tab-1* mutants (for 'touch abnormal') were previously isolated based on their defects in the touch response (L. Carnell, B. Harfe, A. Fire and M. Chalfie, pers. comm.). *Drosophila* Bsh has been implicated in the specification of several neuron types in the *Drosophila* optic lobe (**Hasegawa et al., 2013**). A subset of these neurons are GABAergic (**Raghu et al., 2013**), indicating that a function of Bsh-type homeobox genes in the specification of the GABAergic phenotype may be phylogenetically conserved.

## The GATA2/3 ortholog *elt-1* affects GABA identity of D-type motor neurons

To identify additional regulators of GABAergic identity, we systematically examined the function of *C. elegans* orthologs of genes known to regulate GABAergic identity in the CNS of vertebrates. We examined possible functions of such *C. elegans* orthologs by examining their expression and mutant phenotypes and found a striking example of conserved function. The vertebrate GATA2 and GATA3

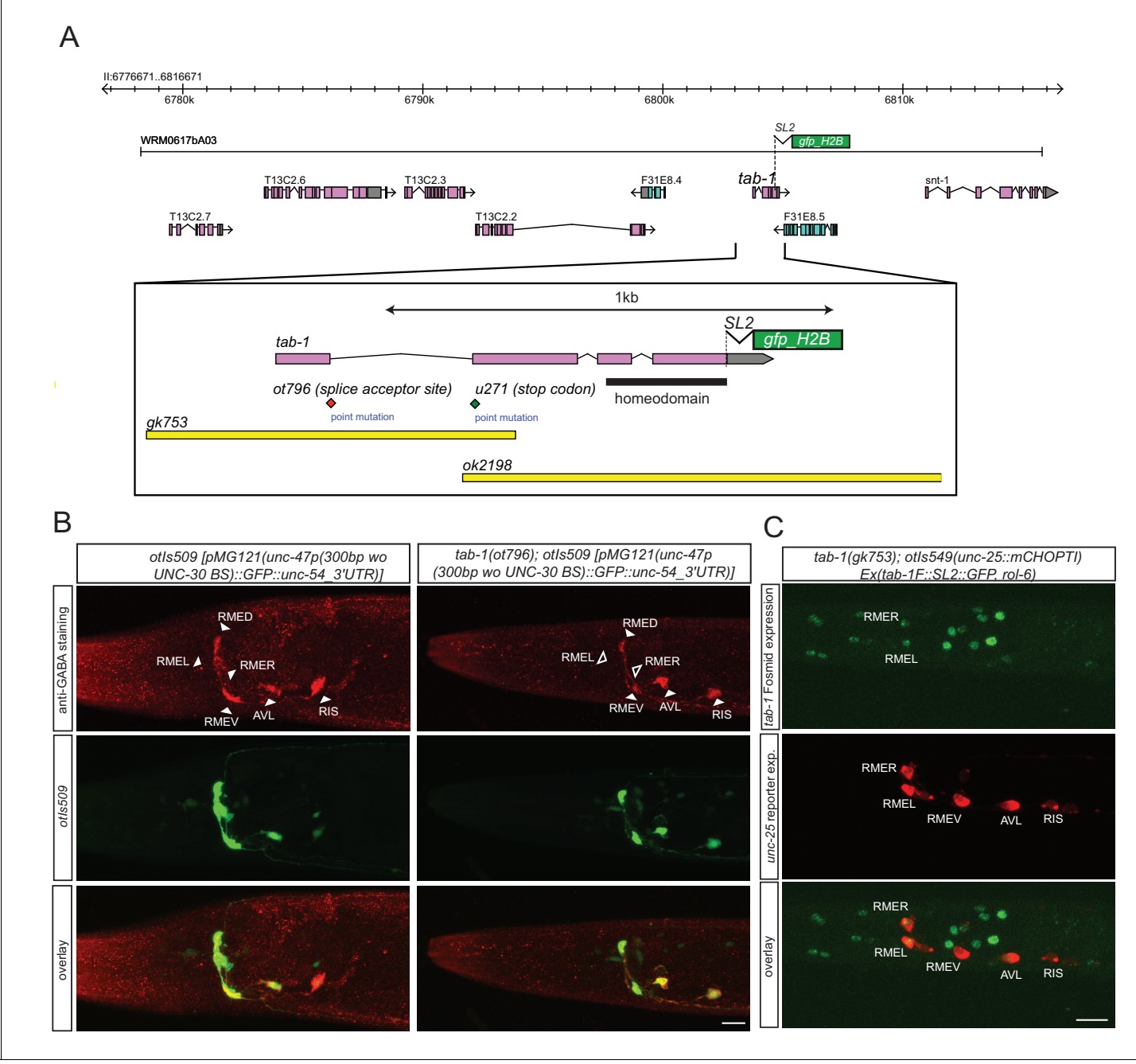

**Figure 6.** *tab-1* is a homeodomain protein cooperating with *nhr-67* to specify RMEL/R GABAergic identity. (**A**) *tab-1* locus, alleles and fosmid reporter. (**B**) *tab-1* affects in RMEL/R *unc-47* reporter expression (in green) and anti-GABA staining (in red). The *unc-47* reporter used has a presumptive *unc-30* binding site deleted, which diminishes the expression of this reporter in D-type neurons. The effects on these and other reporters are quantified in *Table 5* (empty arrow head indicates loss of expression). (**C**) Expression pattern of the rescuing *tab-1* fosmid reporter (in green) in a *tab-1* mutant background with a *unc-25/GAD* gene reporter (in red). (**B–C**) (Scale bar = 10μm)

The following figure supplement is available for figure 6:

**Figure supplement 1.** Whole genome sequencing data for *tab-1* mutant identification.

**Table 5.** Genetic characterization of *tab-1* function in the RME neurons. Expression in RME can not always be unambiguously assigned to the dorsal, ventral, left or right RME neuron, hence only the total number of RME neurons was counted. *otEx6804-6806* are *tab-1*-rescuing arrays (see Material and methods).

| marker | Genotype | Expression observed in: | | | | | |
|--------|----------|-------------------------|--------|--------|--------|--------|----|
| | | 4 RMEs | 3 RMEs | 2 RMEs | 1 RME | 0 RME | n |
| *unc-47^{prom}::gfp* (*otIs509*) | wild type | 100% | 0% | 0% | 0% | 0% | 31 |
| | *ot796* | 0% | 26.7% | 70% | 3.3% | 0% | 30 |
| *unc-47^{prom}::gfp* (*oxIs12*) | wild type | 100% | 0% | 0% | 0% | 0% | 30 |
| | *gk753* | 16.7% | 30% | 53.3% | 3.3% | 0% | 30 |
| *unc-25^{prom}::gfp* (*otIs549*) | wild type | 100% | 0% | 0% | 0% | 0% | 30 |
| | *ot796/+* | 100% | 0% | 0% | 0% | 0% | 22 |
| | *ot796* | 0% | 23.3% | 76.7% | 0% | 0% | 30 |
| | *gk753* | 0% | 20% | 80% | 0% | 0% | 30 |
| | *ot796 otIs549/+* | 6.7% | 36.7% | 56.6% | 0% | 0% | 30 |
| | *ot796/ok2198* | 0% | 42.8% | 57.2% | 0% | 0% | 28 |
| | *ot796/u271* | 3.3% | 36.7% | 60% | 0% | 0% | 30 |
| | *ot796/gk753* | 3.3% | 36.7% | 60% | 0% | 0% | 30 |
| *unc-46^{prom}::gfp*(*otIs575*) | wild type | 93.3% | 6.7% | 0% | 0% | 0% | 30 |
| | *gk753* | 13.3% | 36.7% | 46.7% | 3.3% | 0% | 30 |
| anti-GABA staining | wild type | 70% | 16.7% | 10% | 0% | 3.3% | 30 |
| | *ot796* | 10% | 20% | 63.4% | 3.3% | 3.3% | 30 |
| *tab-1(gk753); otIs549; Ex[tab-1(+),ttx-3::gfp]* | without *otEx6804* | 13.3% | 16.7% | 70% | 0% | 0% | 30 |
| | with *otEx6804* | 80% | 8% | 12% | 0% | 0% | 25 |
| | without *otEx6805* | 5% | 25% | 65% | 0% | 5% | 20 |
| | with *otEx6805* | 41.9% | 9.7% | 48.4% | 0% | 0% | 31 |
| | without *otEx6806* | 5.6% | 16.7% | 77.8% | 0% | 0% | 18 |
| | with *otEx6806* | 66.7% | 10% | 23.3% | 0% | 0% | 30 |

transcription factors operate as selector genes of GABAergic identity in several distinct regions of the vertebrate CNS, including the spinal cord, midbrain, forebrain and hindbrain (*Achim et al., 2014*; *Joshi et al., 2009*; *Kala et al., 2009*; *Lahti et al., 2016*; *Yang et al., 2010*). GATA2 may act transiently after the generation of GABAergic neurons and then pass on its function to the GATA3 paralog. The sole *C. elegans* ortholog of vertebrate GATA2/3 is the *elt-1* gene (*Gillis et al., 2008*), which controls early hypodermal fate patterning (*Page et al., 1997*). We found that an *elt-1* fosmid-based reporter gene is expressed in all D-type motor neurons throughout their lifetime, but not in any other GABA-positive neuron (*Figure 7A and B*). Since *elt-1* mutants display early embryonic lethality, we conducted a genetic mosaic analysis to examine the effect of loss of *elt-1*, function on D-type motor neurons, some of which generated only post-embryonically (the VD MNs). We balanced *elt-1* null mutants with a fosmid that contains the *elt-1* locus and an *unc-47* reporter to assess the loss of the extrachromosomal array in GABAergic D-type neurons. We found that live animals that lost the rescuing array show defects in *unc-25/GAD, unc-47/VGAT* and *unc-46/LAMP* expression in D-type neurons (*Figure 7C*). The loss of expression of GABA markers is not a reflection of loss of the cells, since expression of *unc-30*, the presumptive regulatory co-factor for *elt-1* is still normally expressed in the D-type neurons of *elt-1* mutants (*Figure 7D*).

Two other vertebrate genes controlling GABAergic identity in the vertebrate CNS (Tal1/2 and Lbx; *Achim et al., 2014*) have no *C. elegans* ortholog. The *C. elegans* orthologs of two prominent other regulators of GABAergic neuron identity in vertebrates, the Dlx genes (*ceh-43* in *C. elegans*), and Ptf1a (*hlh-13* in *C. elegans*) have no role in GABAergic identity control since we find that *ceh-43*

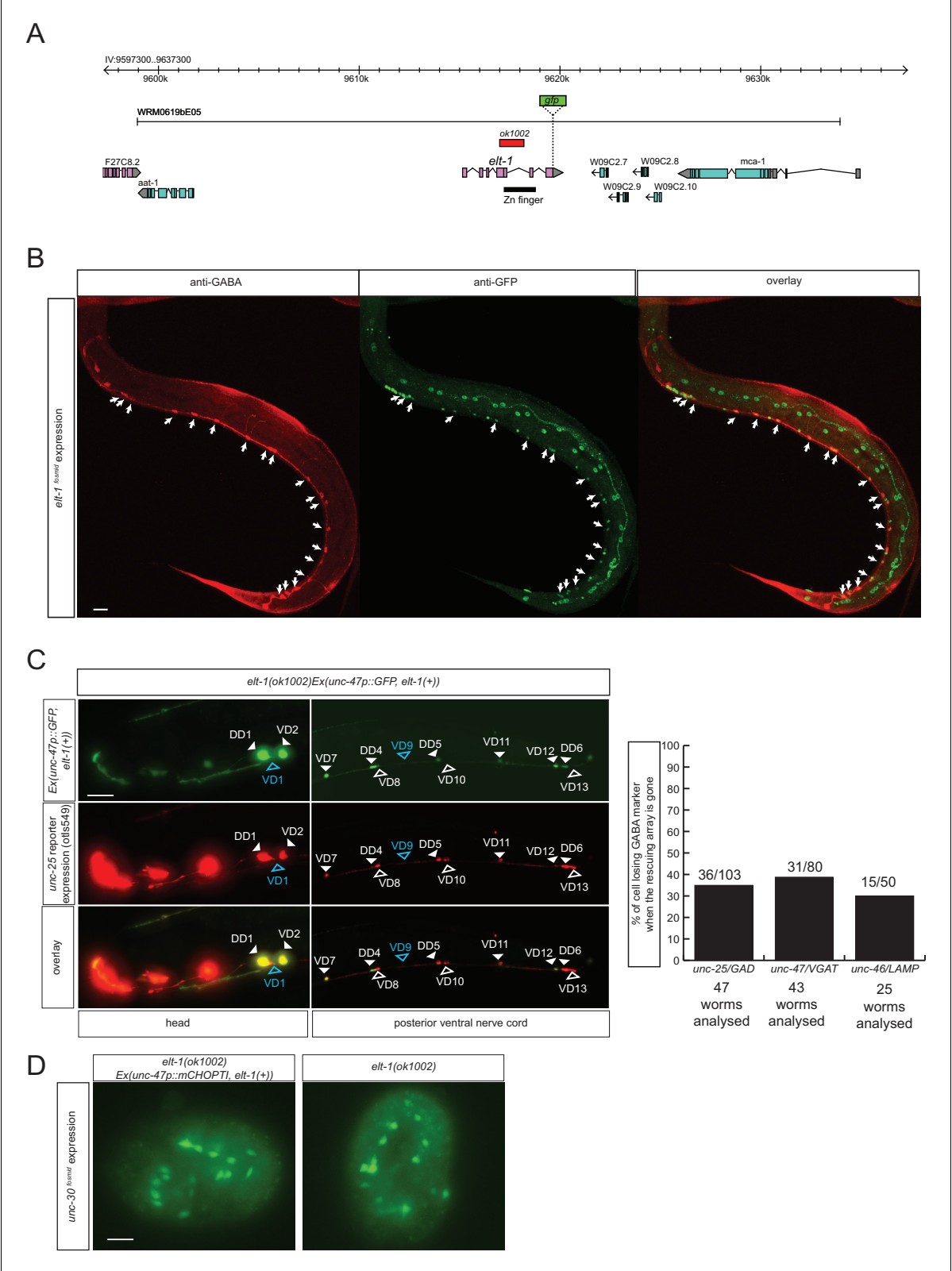

**Figure 7.** The GATA2/3 ortholog *elt-1* controls GABAergic identity of D-type motor neurons. (A) *elt-1* locus and fosmid reporter used for expression analysis. The same, but untagged fosmid was used for mosaic analysis (panel **C**). (B) *elt-1* is expressed in D-type motor neurons. The expression pattern of the *elt-1* fosmid based reporter construct shown after immunostaining with anti-GFP in green and anti-GABA in red (**C**) Mosaic analysis of *elt-1* function. *elt-1* null mutants that carry a rescuing array, as well as an array marker that labels the presence of the array in GABA neurons (*unc-47::gfp*) are

*Figure 7 continued on next page*

*Figure 7 continued*
scored for *unc-25::mChopti (otIs549)* expression. Whenever no *gfp* signal is observed in a D-type motor neuron (empty arrow head), the expression of *unc-25* $^{Prom}$*::mChOpti* is scored (white empty arrow head if present and blue empty arrow head if absent). Bar graph shows similar scoring done for the expression of *unc-47/VGAT* and *unc-46/LAMP*. (**D**) *unc-30* fosmid reporter expression is not affected in *elt-1* mutants, indicating that D-type motor neurons are born in the *elt-1* mutants. (**B–D**) (Scale bar = 10 μm).

is not expressed in mature GABAergic neurons (data not shown) and since *hlh-13* null mutants show no defects in GABA staining (data not shown).

## Homeobox genes controlling the identity of GABA uptake neurons

We next sought to identify factors that control the identity of the GABAergic uptake neurons that we newly identified. The *ceh-14* LIM homeobox gene and the *ceh-17* Prd-type homeobox gene were previously shown to cooperate in the specification of several, peptidergic terminal identity features of the GABA uptake neuron ALA (*Van Buskirk and Sternberg, 2010*). We find that GABA staining of the ALA neurons is abrogated in *ceh-14* and *ceh-17* mutants (*Figure 8A*). Since lack of GABA staining is expected to be due to the failure to express *snf-11/GAT*, we crossed the *snf-11/GAT* fosmid reporter into *ceh-14 and ceh-17* mutants and found its expression to be abrogated in the ALA neuron (*Figure 8A*).

The AVF uptake neuron was previously shown to express the Prd-type homeobox gene *unc-4* (*Miller and Niemeyer, 1995*). We find that *unc-4* mutants do not show GABA staining in AVF and, as a likely reason for the absence of GABA staining, fail to express the *snf-11/GAT* transporter (*Figure 8B*). *unc-3*, a transcription factor controlling the identity of P-cell derived cholinergic neurons, is also expressed in the AVF neurons (derived from P and W), but does not affect anti-GABA staining in AVF (data not shown).

## GABAergic neurotransmitter identity is coupled with the adoption of other identity features

Lastly, we set out to address the question whether factors that control GABAergic identity are committed to only control GABAergic neurotransmitter identity or whether they also control additional identity features of the respective GABAergic neurons. In other words, is the acquisition of GABAergic neurotransmitter identity coupled with the acquisition of other identity features? This appears to indeed be the case if one considers previously published results. Specifically, *ceh-14* and *ceh-17* were previously described to control several identity features of ALA (*Van Buskirk and Sternberg, 2010*) and, as mentioned above, we show here that they also control GABA identity (*Figure 8A* and *Figure 9F*). Similarly, *lim-6*, which specifies GABA identity of RIS and AVL, had previously been found to control several identity aspects of the RIS interneuron, namely expression of two biogenic amine receptors, a glutamate receptor and an Ig domain protein (*Tsalik et al., 2003*) (*Figure 9F*). We found that *lim-6* also controls the expression of a more recently identified RIS marker, *nlr-1*, which encodes a neurexin-like gene (*Haklai-Topper et al., 2011*) and *lim-6* affects expression of the neuropeptide-encoding *flp-22* gene in the AVL neuron (*Figure 9A*). Together with the GABA staining defects of RIS and AVL in *lim-6* mutants described here, this demonstrates that *lim-6* coregulates GABA identity acquisition and other identity features.

We further corroborated the notion of coregulation by examining whether *nhr-67* not only controls GABA identity but also other identity features. We find that in the RIS neurons, *nhr-67* also controls the expression of the 5HT receptor *ser-4* and of *nlr-1* (*Figure 9B*). In the RME neurons, *nhr-67* affects not only GABA identity, but also an expression of the tyramine receptor *ser-2* and the Glu-gated ion channel *avr-15* (*Figure 9B*). *ser-2* and *avr-15* expression is also affected in RMED by *ceh-10*, *ser-2* expression in RMEL/R is affected by *tab-1* and expression of the dynein regulator *bicd-1* in AVF is affected by *unc-4* (*Figure 9C–E*). In conclusion, in most cases examined, regulatory factors that control GABAergic identity features also control other identity features of the neurons examined.

One notable exception to the coregulatory theme appears in the DVB motor neuron. GABA staining is absent in *lim-6* mutants (*Figure 5C*) and *unc-25/GAD* expression is also severely affected (*Figure 5C*) (*Hobert et al., 1999*). However, neither *unc-47::gfp* expression (*Figure 5C*) nor the

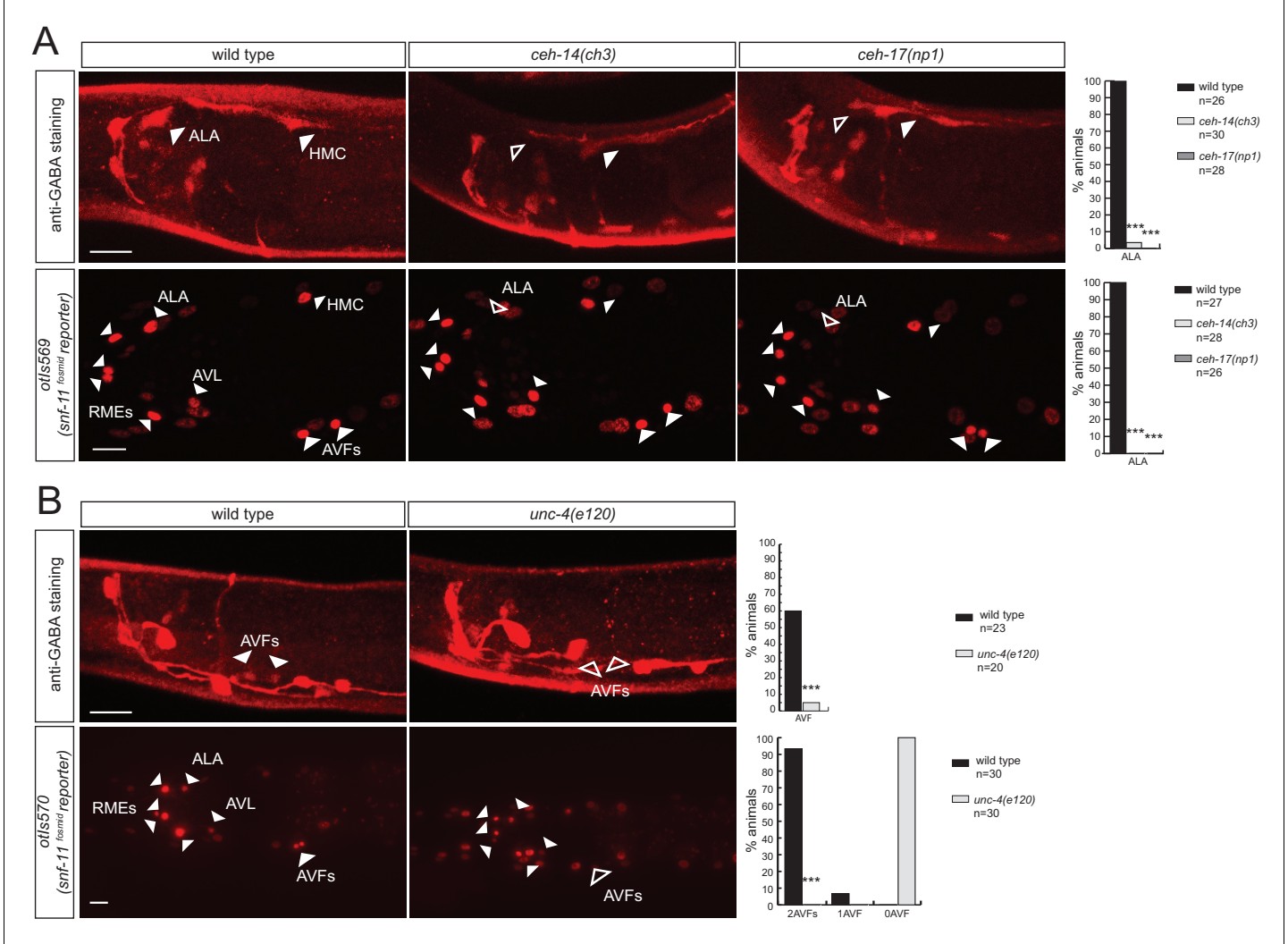

**Figure 8.** Homeobox genes controlling the identity of GABA uptake neurons. (**A**) *ceh-14* and *ceh-17* specify the identity of the newly identified ALA GABAergic neuron, as assessed by GABA staining (upper panels) and *snf-11* fosmid reporter expression (lower panels) (**B**)*unc-4* affects GABA staining (upper panels) and *snf-11* reporter gene expression (lower panels) in the AVF neuron. (**A–C**) plain arrow head indicates a neuron presence and empty arrow head its absence (Scale bar = 10 µm, Fisher exact test, ***p<0.001, compared to wild type; when not noted, not significant; for expression in AVFs statistics were performed only on the category 2AVFs)

expression of two additional DVB markers, *kal-1* and *flp-10,* are affected in *lim-6* null mutants (*Figure 9A*). We note that in the AVL neuron, the effect of *lim-6* on some markers is also very modest, but significantly enhanced if the combinatorial cofactor for *lim-6* in AVL, *nhr-67*, is also removed (*Figure 5D*). We therefore suspect that *lim-6* may act in a partially redundant manner with a cofactor in DVB neuron as well.

## The *unc-42* homeobox gene represses GABA identity in several types of motor- and interneurons

In our search for additional regulators of the GABAergic phenotype, we noted that the normally peptidergic AVK neurons ectopically stain with anti-GABA antibody in animals that lack the *unc-42* homeobox gene (*Figure 10A*). Ectopic GABA staining is accompanied by ectopic *unc-25/GAD, unc-47/VGAT, unc-46/LAMP* and *snf-11/GAT* reporter gene expression (*Figure 10B*). Previous work had shown that *unc-42* induces the expression of multiple aspects of AVK identity (*Wightman et al.,*

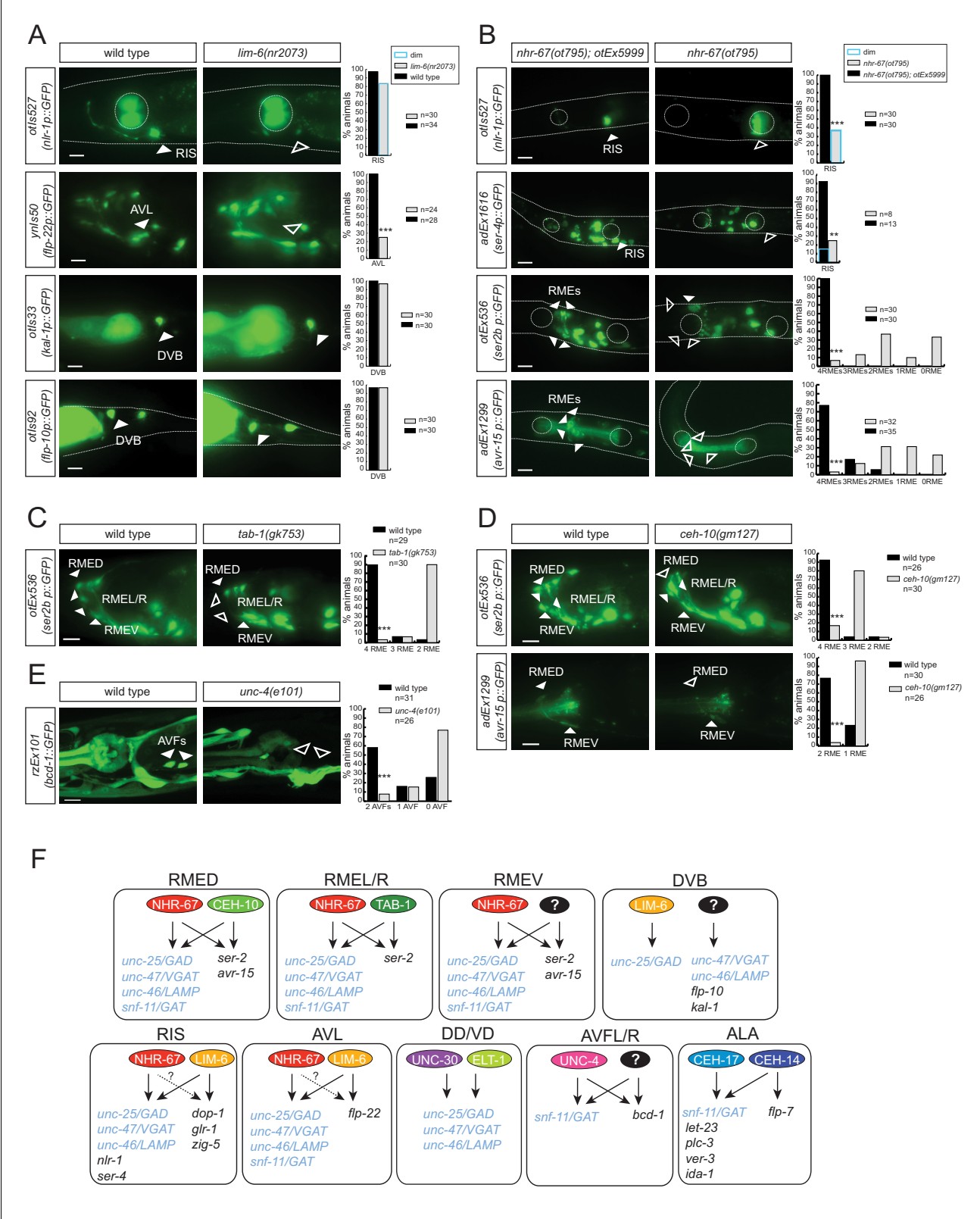

**Figure 9.** Transcription factors controlling GABAergic identity also control other cellular identity markers. (A) *lim-6* controls cell identity markers in AVL and RIS, but not DVB. (B) *nhr-67* controls additional neuron identity markers of GABAergic neurons. *nhr-67(ot795)* animals are not viable and *otEx5999* is an extrachromosomal array that contains copies of the wild-type *nhr-67* locus as well as an array marker (*unc-47^prom^::mChOpti*). Animals that either contain this array or do not contain it were scored for mutant phenotypes at the L1 stage. (C) *tab-1* controls the expression of RMEL/R identity markers.
*Figure 9 continued on next page*

*Figure 9 continued*

(D) *ceh-10* controls the expression of RMED identity markers. (E) *unc-4* controls the expression of the AVF identity marker *bicd-1*. (A–E) plain arrow head indicates a neuron presence and empty arrow head its absence (Scale bar = 10 µm; Fisher exact test, ***p<0.001, compared to wild type; when not noted, not significant; for expression in RMEs or AVFs statistics were performed only on the category 4RMEs or 2AVFs) (F) Summary of known gene regulatory programs in GABA producing neurons (GABA uptake neurons are not shown). '?' indicates the regulatory relationship is just extrapolated from the effect of the transcription factor on GABA pathway genes. Already previously reported were the effect of *lim-6* on non-GABA pathway genes in RIS and AVL, on *unc-25* in DVB (*Hobert et al., 1999*; *Tsalik et al., 2003*), the effect of *nhr-67* on unc-47 (*Sarin et al., 2009*), the effect of *unc-30* on *unc-25* and *unc-47* (*Eastman et al., 1999*), the effect of *ceh-14/ceh-17* on *let-23, plc-3, ver-3, ida-1* and *flp-7* (*Van Buskirk and Sternberg, 2010*).

*2005*). It therefore appears that *unc-42* promotes the peptidergic identity of AVK and suppresses an alternative GABAergic differentiation program.

*unc-42* is also expressed in a cluster of cholinergic motor neurons in the ventral ganglion (*Baran et al., 1999*; *Pereira et al., 2015*). We observed ectopic GABA staining, as well as *unc-25* and *snf-11* reporter gene expression in this region in *unc-42* mutants (*Figure 10*), indicating that *unc-42* may suppress GABAergic differentiation programs in cholinergic motor neurons as well; based on position, the best candidates for the neurons that convert from cholinergic to GABAergic are the normally *unc-42*-expressing SIBD and SIBV motor neuron pairs (*Figure 10A*). Taken together, the existence of a regulatory factor that suppresses GABAergic identity in several distinct neuron types raises the intriguing possibility that GABAergic identity was more broadly expressed in an ancestral nervous system, but suppressed by the recruitment of a factor that could impose an alternative identity on these neurons.

## Discussion

### A *C. elegans* neurotransmitter atlas

This is the third paper in a trilogy of mapping papers that chart the three main neurotransmitter systems in *C. elegans*, Glu, ACh and GABA (*Pereira et al., 2015*; *Serrano-Saiz et al., 2013*). The maps of the major three fast-acting transmitter systems constitute an atlas of neurotransmitter usage whose breadth is unprecedented in any other nervous system. The atlas is shown in *Figure 11*, all neurons are listed in *Supplementary file 1* and a 3D rendering of this atlas is shown in *Video 1*. In total, a neurotransmitter identity has now been assigned to 104 out of the 118 anatomically defined neuron classes of the worm. 98 of these employ a 'classic' fast-acting neurotransmitter (Glu, GABA, ACh), 6 employ exclusively a monoaminergic transmitter (dopamine, serotonin, octopamine or tyramine). Several of the neurons using a fast-acting transmitter also cotransmit a monoamine. For the 14 neuron classes for which no classic neurotransmitter system has been identified so far two scenarios can be envisioned: (1) some of these neurons may be dedicated to the use of neuropeptides, a notion consistent with a preponderance of dark synaptic vesicles and/or paucity of small synaptic vesicles (e.g. BDU, AVH, AVJ, RID); (2) other neurons do contain plenty of conventional small, clear synaptic vesicles (e.g. ASI, AWA, RIR, RMG) and may use presently unknown transmitter systems (*Hobert, 2013*).

### Usage of GABA throughout the nervous system

Among the most notable aspects of this atlas is the previously discussed broad usage of ACh, employed by 52 sensory, inter- and motor neuron classes (out of a total of 118 classes) (*Pereira et al., 2015*) and the apparent paucity of GABA usage. Only nine neuron classes use GABA for synaptic signaling (based on anti-GABA staining and expression of the vesicular transporter) and of those, only two are pure interneurons (RIS and RIB ring interneurons; since AVB, AVA, AVJ and AVF express no known release machinery, we do not consider them as conventional GABAergic interneurons). Due to their locally restricted projections and connectivity, ring interneurons are the next closest thing to vertebrate GABAergic local inhibitory neurons. Yet only two of the 12 ring interneuron classes (collectively referred to as RI... neurons; *White et al., 1986*) utilize GABA. We note that *Ascaris suum*, a distantly related, parasitic nematode, appears to display a remarkably similar set of GABA positive neurons (*Guastella et al., 1991*).

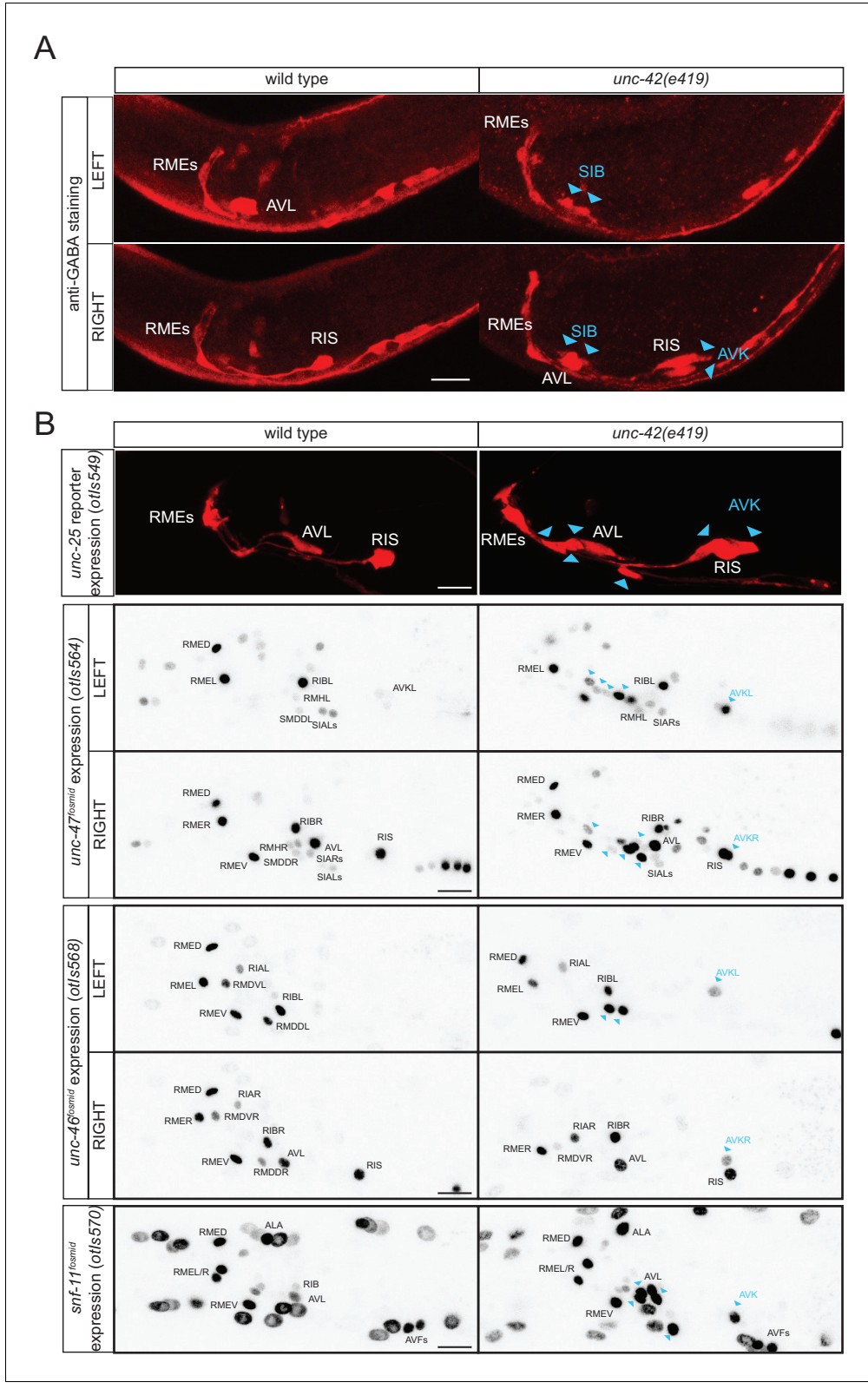

**Figure 10.** *unc-42* represses GABAergic neuron identity. (**A**) SIB and AVK switch to a GABA fate in *unc-42* mutants. Anti-GABA staining in head neurons of wild-type and *unc-42(e419)* mutant. (**B**) GABAergic gene battery is turned on in additional neurons in *unc-42* mutant animals. *unc-25/GAD* gene reporter expression (upper panel) and *unc-47/VGAT, unc-46/LAMP* and *snf-11/GAT* fosmid reporter expression in head neurons of wild-type and *unc-42(e419)*

*Figure 10 continued on next page*

*Figure 10 continued*

mutant animals. Fosmid reporter expression images are color-inverted. (**A–B**) Blue arrow heads point to ectopic expression. (Scale bar = 10 μm)

In the complex male-specific tail circuitry, composed of 20 anatomically distinct neuron classes (*Jarrell et al., 2012*), GABA is also very sparsely used. Only one out of the 20 neuron classes, the EF neuron class, constitutes a GABA neuron with conventional synthesis and release machinery. EF neurons, which project axons into the nerve ring, mostly innervate sex-shared circuitry, and, therefore, GABA may not to be used for communication among male-specific neurons in the tail at all.

The paucity of GABA usage contrasts the much broader usage of GABA in the vertebrate CNS, in which 30–40% of all synapses contain GABA (*Docherty et al., 1985*). However, the apparent paucity of GABA usage in *C. elegans* is no reflection of the paucity of inhibitory neurotransmission in *C. elegans*. First, due to the existence of ACh and Glu-gated chloride channels (*Dent et al., 2000*; *Hobert, 2013*; *Putrenko et al., 2005*), GABA is not the only inhibitory neurotransmitter in *C. elegans*. Second, while only relatively few neurons are GABA positive, a much larger number of neurons may be responsive to GABA. This can be inferred from the expression patterns of ionotropic GABA$_A$-type neurotransmitter receptors, which extend beyond the limited number of neurons that are innervated by GABA-positive neurons. Such expression is consistent with wide-spread spillover transmission. However, while spillover transmission definitely does occur in the ventral nerve cord (*Jobson et al., 2015*), it remains to be experimentally tested whether these GABA receptors indeed engage in GABA spillover transmission. Another indication for wide-spread spillover transmission is the notable restriction of expression of the GABA uptake transporter GAT that was already previously noted (*Mullen et al., 2006*). In vertebrates, these transporters are expressed widely throughout the CNS, with most GABA-positive neurons also expressing GAT (GAT1 or GAT3); many postsynaptic targets of vertebrate GABAergic neurons also express GAT (*Conti et al., 2004*; *Swan et al., 1994*; *Zhou and Danbolt, 2013*). In notable contrast, the sole *C. elegans* GAT ortholog SNF-11 is only expressed in a small fraction of the GABA-positive neurons and it is only expressed in two types of GABAergic targets cells (AVL as a target of DVB and body wall muscle as targets of D-type motor neurons)(this paper; (*Mullen et al., 2006*)). We hypothesize that the restricted

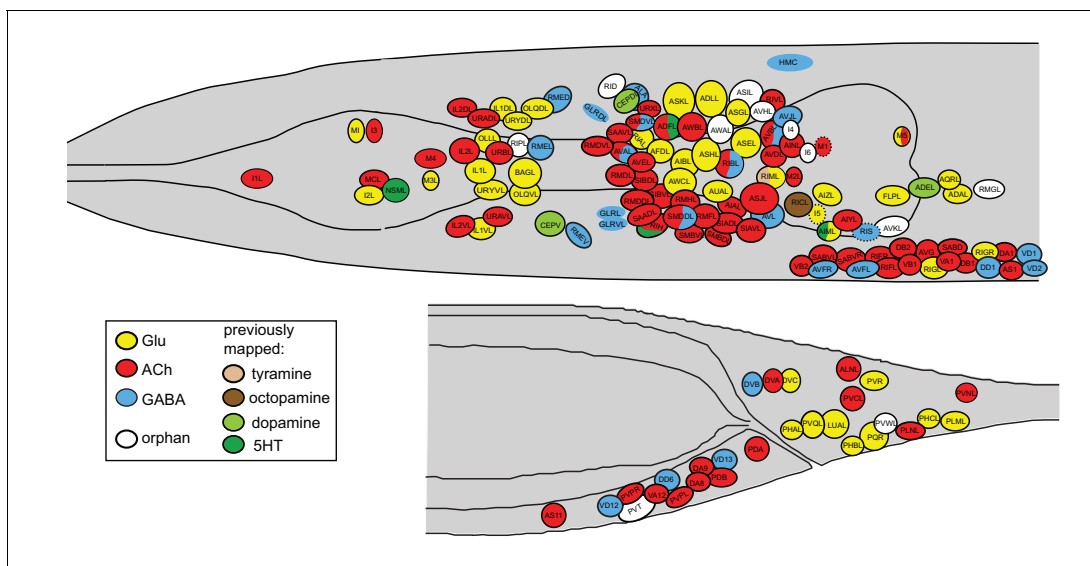

**Figure 11.** Neurotransmitter atlas. The current status of the *C. elegans* neurotransmitter atlas is shown. Only the head and tail of the worm are shown (left view). The Glu and ACh maps come from (*Pereira et al., 2015*; *Serrano-Saiz et al., 2013*) and have been updated here with the GABA neuron analysis. See *Supplementary file 1* for complete list. Dashed circle indicates neurons that are only present on the right side of the animal.

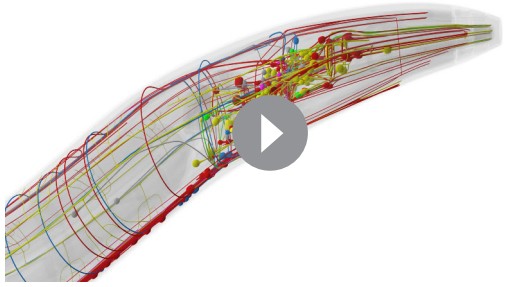

**Video 1.** Flyover movies ACh (red), Glu (mint), GABA (blue), monoaminergic (green) neurons. This movie was generated by Chris Grove.

expression of SNF-11/GAT is a reflection of GABA not being immediately cleared after release, but being able to spill over to signal to non-synaptic targets.

Some of the GABAergic neurons that we newly identified here cotransmit ACh (one of them, the RIB interneuron only expresses a very low level of the ACh synthesizing and transporting machinery) (*Pereira et al., 2015*). ACh/GABA-cotransmitting neurons have been observed in multiple neuron types of the vertebrate CNS as well (*Granger et al., 2016*). Neurons that use two neurotransmitters can, in principle, package both neurotransmitters into the same vesicles or package them separately into spatially segregated presynaptic zones (*Vaaga et al., 2014*). The GABA- and ACh-positive SMD neurons synapse onto two fundamentally distinct cell types – head muscles and a number of distinct inter- and motor neurons (*Table 1*) (*White et al., 1986*) and may differentially segregate ACh and GABA to distinct target synapses.

## GABA uptake: GABA recycling and GABA clearance neurons

Our studies define not only novel GABA-synthesizing and GABA-releasing neurons but also neurons that we term 'GABA uptake neurons'. The GABA-positive nature of these neurons critically depends on the GABA uptake transporter SNF-11/GAT that is expressed in these neurons. Such GABA uptake neurons also exist in *Ascaris suum* (*Guastella and Stretton, 1991*). Based on the expression of the vesicular GABA transporter UNC-47/VGAT, we propose that GABA uptake neurons fall into two categories, 'GABA recycling neurons' and 'GABA clearance neurons'. GABA clearance neurons (the AVF neurons) take up GABA but because these cells do not express the GABA vesicular transporter UNC-47, they do not appear to be capable of re-utilizing GABA, i.e. packaging GABA in synaptic vesicles for re-release (however, we caution that GABA may be released by AVF via non-conventional means, as observed in vertebrates [*Koch and Magnusson, 2009*; *Lee et al., 2010*]). The axon of AVF extends through the nerve ring and along the ventral nerve cord and AVF may therefore clear GABA released from several different GABAergic neuron types. GABA clearance by AVF may control communication between GABA-releasing neurons and their postsynaptic, GABA receptor-expressing targets. AVF may also restrict and spatially define non-synaptic GABA spillover transmission. Whether analogous GABA clearance neurons exist in the vertebrate CNS is as yet unclear, but it is notable that the vertebrate CNS does contain neurons that do not synthesize GABA but take it up via GAT (*Conti et al., 2004*; *Swan et al., 1994*). However, it is generally assumed that these neurons are postsynaptic to GABAergic neurons and hence, that GABA uptake occurs at the synapse. In contrast, AVF is not a synaptic target of GABAergic neurons.

Another potential type of GABA uptake neurons not only expresses the GABA uptake transporter GAT, but also expresses the UNC-47/VGAT vesicular transporter. We speculate that these neurons are possible 'GABA recycling neurons' that synaptically release GABA after uptake. The ALA neuron class falls into this category. ALA, which extends two processes in the nerve ring, may take up GABA released from any of the GABA-releasing neurons that also extend processes in the nerve ring (*Figure 1B*), with SMD neurons being the best candidates due to the direct adjacency of their processes. While we do not have direct evidence that ALA then re-releases GABA, ALA has previously been shown to inhibit the activity of the synaptically connected AVE command interneurons to control locomotory behavior (*Fry et al., 2014*). Since ALA does not express any other known fast-acting transmitter system, we posit that this inhibitory activity is mediated by GABA released by ALA and perceived by ionotropic GABA receptors expressed in AVE (*Table 4*). GABA uptake by ALA, followed by GABA release may serve to coordinate the activity of GABAergic neurons in the nerve ring (e.g. SMDs) with ALA and AVE and eventually locomotory activity. While further studies are required to test the concept of 'GABA recycling neurons' in *C. elegans*, we note an interesting precedent of GABA recycling in the vertebrate CNS. Midbrain dopaminergic neurons do not synthesize GABA,

but take it up via the GABA transporters GAT1 and GAT4 and then release GABA to inhibit postsynaptic neurons (*Tritsch et al., 2012*, *2014*).

We also discovered a group of unusual GABA-positive neurons, the AVA, AVB, AVJ head interneurons. These cells express low but clearly detectable levels of GABA and require *unc-25/GAD* for their GABA staining. However, these neurons fail to express the vesicular transporter UNC-47 or the GABA uptake transporter SNF-11 (which, in other systems is sometimes used to release GABA, rather than take up GABA). Their GABA-positive nature does not depend on SNF-11 and, under the perhaps incorrect assumption that no other transporter can uptake GABA, these neurons therefore do not serve to clear GABA. Since these neurons do not express known transporters to release GABA, they may either employ non-conventional release mechanisms (*Koch and Magnusson, 2009*; *Lee et al., 2010*) or may simply not engage in GABA signaling at all.

In conclusion, the extent to which GABA recycling or GABA clearance neurons exist in the vertebrate CNS remains unclear but we have used here the simplicity of the *C. elegans* nervous system to precisely define the set of GABA synthesizing and GABA uptake neurons.

## Remarkable conservation of GABA receptor organization

The previous genome sequence analysis has revealed a remarkable conservation in the organization of GABA receptor genes (*Darlison et al., 2005*; *Tsang et al., 2007*). In all species examined, GABA receptor genes are located in genomic clusters. Based on the patterns of clustering, the existence of an 'ancient' cluster of GABA receptors has been proposed which then duplicated multiple times in vertebrates. One component of this cluster is the genes that code for alpha and beta subunit GABA receptors, the two obligatory subunits of a functional GABA receptor. Intriguingly, these two subunits are always located in a head-to-head manner throughout all animal genomes (*Darlison et al., 2005*; *Tsang et al., 2007*), suggesting that alpha and beta genes share the same regulatory elements to be expressed in the same neuron types. However, this notion has not been examined in a nervous system-wide manner with single cell resolution. Our reporter gene analysis provides exactly that confirmation. Based on fosmid-based transgenes we find that the head-to-head organized *gab-1* and *lgc-37* genes are indeed co-expressed. Reporter gene fusions in which we examined the regulatory content of the intergenic region of *gab-1* and *lgc-37* in each orientation showed that this co-expression is indeed ensured by the same *cis*-regulatory control elements essentially operating in two different orientations. In the context of neurotransmission, we are only aware of one other case of such remarkably conserved genomic linkage; in this case, the enzyme for ACh synthesis and the vesicular transporter of ACh are located adjacent to one another and share the same first exon (*Alfonso et al., 1994*).

## GABA in non-neuronal cells

Apart from the easily explicable detection of GABA in muscle cells, the targets of the largest class of GABAergic motor neurons, we detected GABA in two intriguing and unexpected non-neuronal cell types, the unusual hmc and the glia-like GLR cells. Both cell types may operate in GABA clearance. In vertebrates, some glial cell types are thought to employ GABA as a 'gliotransmitter', releasing GABA via a reversal of the plasma membrane GABA transporter GAT-1 to signal to neurons (*Barakat and Bordey, 2002*; *Koch and Magnusson, 2009*; *Yoon and Lee, 2014*). The GLR cells indeed express the *C. elegans* ortholog of the GAT-1 GABA transporter (SNF-11) and it will be intriguing to test whether the GLRs indeed also engage in active GABA signaling.

## Regulation of the GABA phenotype

We used the map of GABA-positive neurons as an entry point to study how neurons acquire their GABAergic phenotype (*Figure 9F*). We built on previous work that implicated a few factors in controlling GABAergic features, extending the mutant analysis of these factors and identifying novel combinatorial codes of transcription factors that define GABAergic identity. We also uncovered factors that define the identity of GABA clearance and recycling neurons. Our work corroborates and extends a number of previously developed themes and concepts:

## Combinatorial transcription factor codes

Transcription factors that specify the GABAergic phenotype act in neuron-type specific combinations (*Figure 9F*). Each GABAergic neuron type uses its own specific combination of regulators and, hence, there is no commonly employed inducer of GABAergic identity. This conclusion could already be derived from previous work (*Hobert et al., 1999*; *Jin et al., 1994*) and we extend this conclusion here by defining the nature of several of the combinatorial transcription factor codes. Nevertheless, there is a notable reiterative use of two different regulators, *nhr-67* (RME, AVL, RIS) and *lim-6* (AVL, RIS, DVB) in specifying GABA identity in different cellular contexts. The neurons that are specified by *nhr-67* and *lim-6* are synaptically connected (*White et al., 1986*) and perhaps these factors may have a role in circuit assembly as well, as previously suggested for other 'circuit-associated transcription factors' (*Pereira et al., 2015*).

## Preponderance of homeobox genes

The majority of regulators of neuronal identity (of GABA, but also Glu and ACh neurons) are encoded by homeobox genes. Those that are not (*nhr-67* and *elt-1*) cooperate with homeobox genes (*Figure 9F*). This is notable in light of the fact that only ~10% of all transcription factors encoded by the *C. elegans* genome are of the homeodomain type (*Reece-Hoyes et al., 2005*). This observation suggests that homeobox genes may have been recruited into neuronal specification early in evolution and that these homeobox-mediated blueprints then duplicated and diversified to generate more and more complex nervous systems.

## Phylogenetic conservation

Vertebrate GATA2/3 factors are postmitotic selectors of GABAergic identity in multiple distinct GABAergic neuron types (*Achim et al., 2014*; *Joshi et al., 2009*; *Kala et al., 2009*; *Lahti et al., 2016*; *Yang et al., 2010*). We found that its *C. elegans* ortholog *elt-1* also specifies a GABAergic neuron identity, apparently in conjunction with the *unc-30/Pitx* gene. Remarkably, a population of GABAergic neurons in the CNS also co-expresses the mouse orthologs of *elt-1* and *unc-30* (*Kala et al., 2009*). All other factors we identified in *C. elegans* have vertebrate orthologs as well and according to the Allen Brain Atlas (*Sunkin et al., 2013*) are expressed in the adult CNS. It will need to be tested whether these orthologs are expressed and function in GABAergic neurons. Notably, however, the *C. elegans* ortholog of the Dlx genes, well characterized selectors of GABAergic identity in the anterior forebrain of the mouse (*Achim et al., 2014*), does not appear to be involved in GABAergic neuron differentiation in *C. elegans*.

## Coupling of GABAergic identity with other identity features

The decision to acquire a GABA-positive phenotype is coupled to the acquisition of other terminal identity features. This is evidenced by the genetic removal of transcriptional regulators described here; such loss does not only result in the loss of GABAergic features, but also the loss of expression of other genes that define mature neuronal features, such as neuropeptides, ion channels, monoaminergic transmitter receptors and others. Transcription factors that control the expression of distinct terminal identity features have been termed 'terminal selectors' (*Hobert, 2008*) and much of the data shown here support the terminal selector concept. However, there are also exceptions: in the DVB motor neuron, the *lim-6* LIM homeobox gene controls expression of *unc-25/GAD*, but not *unc-47/VGAT*, *unc-46/LAMP* or other identity features. A similar de-coupling of regulation of terminal identity features has been observed in the specification of the serotonergic neuron type NSM (*Zhang et al., 2014*) and in cholinergic command interneurons (*Pereira et al., 2015*). In the case of NSM, the loss of one homeobox gene appears to be compensated for the action of a redundantly acting homeobox gene (*Zhang et al., 2014*).

## Similar neurons with distinct lineage histories are specified by the same terminal selector

As noted by John White more than 30 years ago, the four RME neurons are one example of a class of anatomically similar neurons whose individual class members have very distinct lineage histories (*White, 1985*). We have shown here that the similarity of the four RME neurons is apparently genetically programmed by a shared terminal selector, *nhr-67*, which appears to endow the four RME

neurons its class-defining properties. RME subtype-specific properties (i.e. genes that are expressed only by a subset of the RME neurons) are also controlled by *nhr-67*, but the subtype-specificity is controlled by subtype-specific transcription factors that appear to collaborate with *nhr-67*. We had previously derived a similar conclusion for the *C. elegans* dopaminergic neuron classes, which are also morphologically and molecularly very similar, but display distinct lineage histories; the unifying features of all lineally distinct dopaminergic neurons appear to be specified by the same terminal selector-type transcription factors (*Doitsidou et al., 2013*; *Flames and Hobert, 2009*)

### A system-wide regulatory map of neurotransmitter specification

All of the five conclusions derived here from our analysis of *C. elegans* GABA-positive neuron specification conform with similar conclusions derived from the analysis of the specification mechanisms of *C. elegans* cholinergic neurons (*Pereira et al., 2015*), glutamatergic neurons (*Serrano-Saiz et al., 2013*) and monoaminergic neurons (*Doitsidou et al., 2013*; *Sze et al., 2002*; *Zhang et al., 2014*; *Zheng et al., 2005*). The regulatory mechanisms for all these transmitter systems can be synthesized into a 'regulatory map' of neurotransmitter specification, shown in *Figure 12*. As shown in this figure, a view across different neurotransmitter systems illustrates that the activity of individual terminal selectors of neurotransmitter identities is not confined to specific neurotransmitter systems. For example, the *ceh-14* homeobox gene acts with different homeobox genes to specify GABA identity (ALA; this paper), glutamatergic identity (*Serrano-Saiz et al., 2013*) or cholinergic identity (*Pereira et al., 2015*). This reuse is remarkable if one considers that the four most re-employed transcription factors (*unc-3, unc-42, ceh-14, unc-86*) are involved in specifying the neurotransmitter identity of 46 of the 69 neuron classes for which a neurotransmitter regulatory is known (*Figure 12*). We conclude that the system-wide view of neuronal specification, using distinct neurotransmitter systems, has begun to reveal common organizational principles of neuronal specification (further discussed in *Hobert, 2016a*, *2016b*).

## Materials and methods

### Mutant strains

The *C. elegans* mutant strains used in this study were: *unc-47(e307)*, *unc-25(e156)*, *snf-11(ok156)*, *unc-30(e191)*, *nhr-67(ot795)*; *otEx5999 [nhr-67 fosmid, pMG92(unc-47prom::mChOpti)]*, *lim-6 (nr2073)*, *ceh-10(gm127)*, *ceh-10(gm133)/hT2*, *tab-1(ot796)*, *tab-1(gk753)*, *tab-1(ok2198)*; *tab-1 (u271)*, *ceh-14(ch3)*, *ceh-17(np1)*, *unc-4(e120)*, *elt-1(ok1002) IV/nT1 [qIs51] (IV;V)*, *unc-42(e419)*.

### Transgenic reporter strains

The *unc-47, unc-46, gta-1, snf-11, tab-1, lgc-36, lgc-37* and *gab-1* fosmid reporter constructs were generated using λ-Red-mediated recombineering in bacteria as previously described (*Tursun et al., 2009*). For the *unc-47, unc-46, unc-25, snf-11* and *gta-1* fosmid reporters, an SL2 spliced, nuclear-localized mChOpti::H2B sequence was engineered right after the stop codon of the locus (mChOpti = codon optimized mCherry). For the *tab-1, lgc-36, lgc-37* and *gab-1* fosmid reporter, an SL2 spliced, nuclear-localized YFP::H2B sequence was engineered right after the stop codon of the locus.

With the exception of the *tab-1* fosmid, fosmid DNA was generally injected at 15 ng/μL into a *pha-1(e2123)* mutant strain with *pha-1(+)* as co-injection marker (*Granato et al., 1994*) for *unc-47, unc-46, gta-1* and *snf-11* and with *pha-1(+) ttx-3::mChOpti* as co-injection marker for *lgc-36, lgc-37* and *gab-1*. The *tab-1* fosmid reporter DNA was injected at 15 ng/μL into *tab-1(gk753) otIs549* mutant strain with *rol-6(RF4)* as co-injection marker. Some of the resulting transgenes were chromosomally integrated. Resulting transgenes are: *otIs564* for *unc-47, otIs568* for *unc-46, otIs569* and *otIs570* for *snf-11 (otIs570* was used for most of the experiments unless otherwise specified), *otEx6746* for *gta-1, otEx6747* for *tab-1, otEx6798* for *lgc-36, otEx6799* for *lgc-37* and *otEx6800* for *gab-1*.

Rescuing experiment was performed by injecting a PCR fragment of the 5 kb intergenic region of *tab-1* with *ttx-3::gfp* as co-injection marker in *tab-1(gk753) otIs549*; three independent lines were generated and analyzed: *otEx6804, otEx6805* and *otEx6806*.

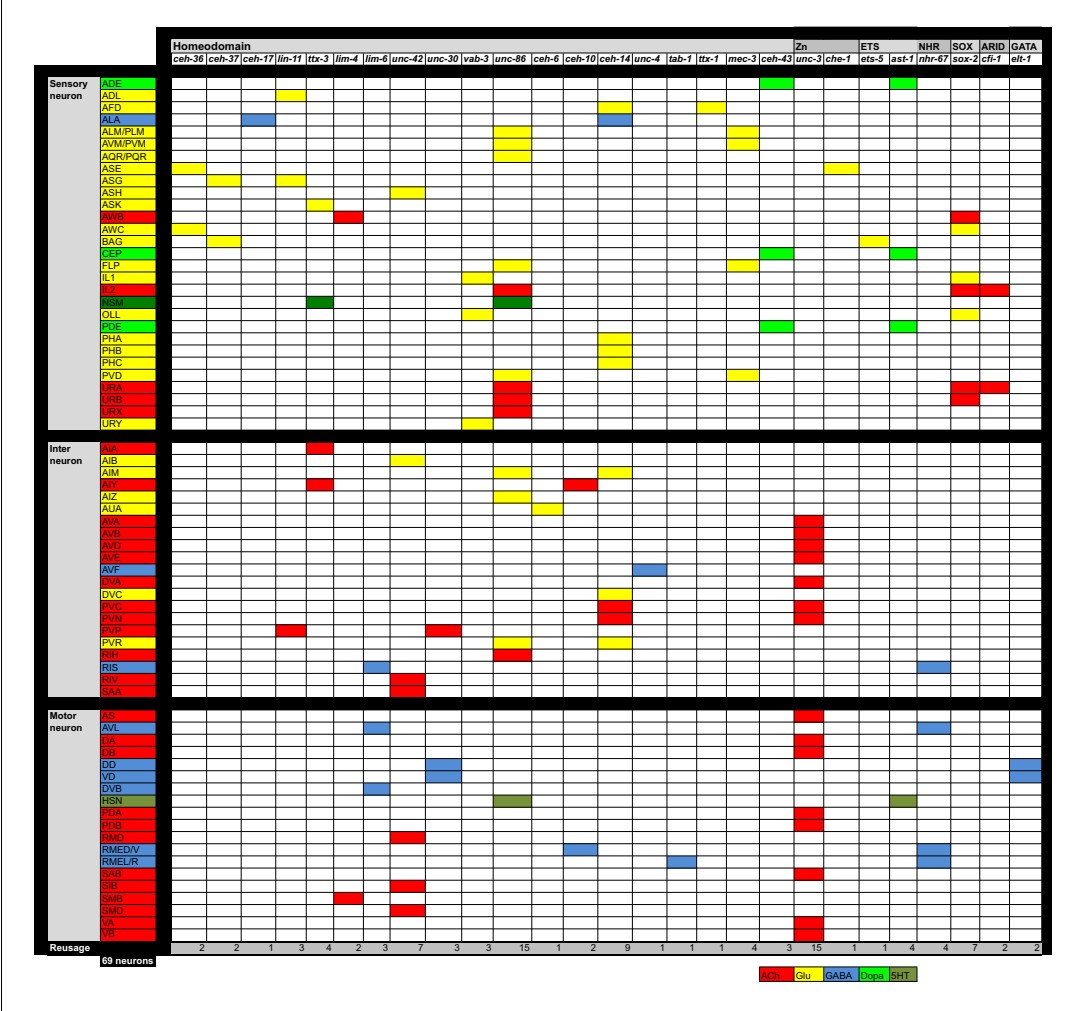

**Figure 12.** Regulatory map of neurotransmitter specification. All neurons are shown for which neurotransmitter identity has been determined and for which a regulator for the respective neurotransmitter identity has been identified. Colored boxes indicate cells in which the indicated transcription factor affects neurotransmitter specification. Data are from this paper for GABA and from following papers for additional neurons: (*Alqadah et al., 2015*; *Altun-Gultekin et al., 2001*; *Doitsidou et al., 2013*; *Flames and Hobert, 2009*; *Kratsios et al., 2015, 2012*; *Pereira et al., 2015*; *Serrano-Saiz et al., 2013*; *Sze et al., 2002*; *Vidal et al., 2015*; *Zhang et al., 2014*)

The following reporter strains were generated for this study by injecting the PCR product from pPD95.75 plasmids containing the upstream region of the gene at 5 ng/µL into a *pha-1(e2123)* mutant strain with *pha-1(+)* as co-injection marker: *nlr-1^prom^::gfp (otIs527*, 150bp upstream the ATG); *unc-47^prom^::gfp (otIs509*, 300bp upstream the ATG with a deleted *unc-30* binding site to reduce expression in D-type motor neurons); *unc-25::mChOpti (otIs549*, 5.1 kb upstream the 4th exon); *unc-25::gfp (otIs514*, 7 kb upstream the 6th exon); *unc-46^prom^::gfp (otIs575, 234bp upstream the ATG)*.

For the GABA receptor reporters, the respective promoter region was cloned in front of the *gfp::unc-54-3'UTR*. For *lgc-38*, 3.5 kb upstream the 3rd exon was used and 3.9 kb upstream the ATG; for *lgc-37*, 5 kb upstream the ATG was used; for *gab-1*, the exact same 5 kb but reverse was used. The *lgc-38(3.5 kb)* reporter strain is from (*Wenick and Hobert, 2004*). *lgc-37* and *gab-1* reporter strains were made by injecting a PCR product at 5 ng/µL into a *pha-1(e2123)* mutant strain with *pha-1(+)* as co-injection marker (resulting transgenes: *otEx6801* for *lgc-38(3.9 kb)*, *otEx6802* for *lgc-37* and *otEx6803* for *gab-1*).

The fosmid WRM0619bE05 (elt-1(+)) was injected in the mutant strain *elt-1(ok1002) IV/nT1 [qIs51] (IV;V)* with either *unc-47^prom ::gfp (otEx6751)* or *unc-47^prom ::mChOpti (otEx6750)* as co-injection marker. After lines were generated, worms carrying the array were singled. After three days, plates containing 100% worms with the array were isolated and used for subsequent analysis of *elt-1*.

The following additional, and previously described neuronal markers were used in the study: *unc-47^prom::gfp (oxIs12)*, *unc-47^prom::gfp (otIs348)*, *ser-4 ^prom::gfp (adEx1616)*, *ser-2b ^prom::gfp (otEx536)*, *avr-15 ^prom::gfp (adEx1299)*, *flp-22 ^prom::gfp (ynIs50)*, *kal-1 ^prom::gfp (otIs33)*, *flp-10 ^prom::gfp (otIs92)*, *wgIs354 [elt-1::TY1::EGFP::3xFLAG + unc-119(+)]*, *rab-3 ::NLS ::tagRFP (otIs355)*, *cho-1^fosmid ::SL2 :: YFP ::H2B (otIs354)*, *eat-4^fosmid ::SL2 ::YFP ::H2B (otIs388)*, *cho-1^fosmid ::SL2 ::mChOpti ::H2B (otIs544)*, *eat-4^fosmid ::SL2 ::mChOpti ::H2B (otIs518)*, *wgIs395 [unc-30::TY1::EGFP::3xFLAG + unc-119(+)]*, *lim-6 ^rescuing fragment ::gfp (otIs157)*, *nhr-67^fosmid ::mChOpti (otEx3362)*, *bicd-1:: gfp(rzEx101)*

## Genome engineering

*Generation of the nhr-67(ot795) deletion allele:* The *nhr-67* null allele *ot795* was generated by transposon excision (MosDEL) as previously described (*Frøkjaer-Jensen et al., 2010*), using ttTi43980, a *Mos1* insertion in the first intron of *nhr-67* kindly provided by the NemaGENETAG Consortium. The resulting *nhr-67(ot795)* allele contains a 4.5 kb deletion, including the whole *nhr-67* coding region except for the 1st exon, as verified by PCR analysis and sequencing.

*Generation of the unc-25(ot867[unc-25::SL2::gfp]) gfp knock-in allele:* the *gfp* knock-in allele into the *unc-25* locus was generated using CRISPR/Cas9-triggered homologous recombination alongside with a self-excising cassette (SEC) for drug selection as previously described (*Dickinson et al., 2015*). The resulting *unc-25(ot867[unc-25::SL2::gfp])* allele contains an SL2::1xNLS::GFP-3xFLAG:: H2B right after the 2nd predicted STOP codon.

## GABA staining

A previously described GABA staining protocol (*McIntire et al., 1993b*) was modified in the following manner. L4/young adult hermaphrodites or males were fixed for 15 min (as opposed to 24 hr) at 4°C in PBS (137 mM NaCl, 2.7 mM KCl, 10 mM $Na_2HPO_4$, 2 mM $KH_2PO_4$), 4% paraformaldehyde/ 2.5% glutaraldehyde fixative (as opposed to 4% paraformaldehyde/ 1% glutaraldehyde fixative). After being washed three to four times in PBS/0.5% Triton X-100, the worms were rocked gently for 18 hr at 37°C in a solution of 5% β-mercaptoethanol, 1% Triton X-100 in 0.1 M Tris-HCl(pH 7.5) (as opposed to 0.125 M Tris-HCl(pH 6.9)). The worms were washed four times in 1% Triton X-100/0.1 M Tris-HCl(pH7.5) and one time in 1 mM $CaCl_2$/1% Triton X-100/0.1 M Tris-HCl(pH7.5). A worm pellet of 20–50 µL was shaken vigorously in 1 mL of 1 mM $CaCl_2$/1% Triton X-100/0.1 M Tris-HCl(pH7.5) and 1 mg/mL of collagenase type IV (C5138, Sigma) for 30 min. The worms were then washed three times in PBS/0.5% Triton X-100. An extra step was added in order to quench the autofluorescence due to the glutaraldehyde: the worms were incubated for one hour at 4°C in a freshly made solution of PBS and 1 mg/mL of $NaBH_4$ (Sigma, 71321).

Samples were blocked for 30 min at room temperature with 0.2% gelatine from fish (Sigma). Anti-GABA antibodies (abcam, ab17413) were used at a 1:250 dilution. For double labelling, anti-GFP (Thermo Fisher, A10262) or anti-RFP (MBL PM005) was used at a 1:1000 and 1:500 dilution respectively. Incubations were done overnight at 4°C. Secondary antibodies included Alexa-488-labelled-goat-anti-chicken (Invitrogen, A11039), Alexa-488-labelled-goat-anti-guinea pig (life, A11073), Alexa-555-labelled-goat-anti-guinea pig (life, A21435) or Alexa-594-labelled-donkey-anti-rabbit (Invitrogen, A21207).

## EMS screen and *tab-1* cloning

An ethyl methanesulfonate (EMS) mutagenesis was performed on the reporter strain *otIs509* driving GFP expression in the 26 'classic GABA neurons'. 6762 haploid genomes were screened for abnormal expression of GFP. A mutant lacking *gfp* expression in RMEL/R (*ot796*) was isolated. After checking for the recessivity of the allele, *ot796* was crossed into the Hawaiian strain and 51 F2s missing RMEL/R were isolated and prepared for Whole genome sequencing as described in (*Doitsidou et al., 2010*). The results were then analyzed employing the CloudMap data analysis pipeline (*Minevich et al., 2012*). Complementation tests between *ot796* and three alleles of *tab-1* (*u271, ok2198* and *gk753*) confirmed that *ot796* is an allele of *tab-1*.

## Microscopy

Worms were anesthetized using 100 mM of sodium azide ($NaN_3$) and mounted on 5% agarose on glass slides. All images (except *Figure 5B,C,E*, *Figure 7C,D* and *Figure 9*) were acquired using a Zeiss confocal microscope (LSM880). Several z-stack images (each ~0.45 μm thick) were acquired with the ZEN software. Representative images are shown following orthogonal projection of 2–10 z-stacks. Images shown in *Figure 5B,C,E*, *Figure 7C,D* and *Figure 9* were taken using an automated fluorescence microscope (Zeiss, AXIOPlan 2). Acquisition of several z-stack images (each ~0,5 μm thick) was performed with the Micro-Manager software (Version 3.1). Representative images are shown following max-projection of 2–10 z-stacks using the maximum intensity projection type. Image reconstruction was performed using ImageJ software (*Schneider et al., 2012*).

## Acknowledgements

We thank Q Chen for generating transgenic strains, C Grove at Wormbase for generating the fly-over movie of neurotransmitter identity, L Carnell, B Harfe, A Fire and M Chalfie for communicating unpublished results on *tab-1*, I Topalidou and M Chalfie for *gab-1* and *lgc-37* reporter plasmids, J Ghergurovich for help with genetic screening and the original isolation of the *tab-1* allele, E Serrano-Saiz and L Pereira for advice on cell identifications and comments on the manuscript, K Howell for comments on the manuscript, M Chalfie and J Rand for extensive discussions and comments on the manuscript, J Huang for discussions on vertebrate GABAergic neurons, S Cook for analysis EM cross sections, the Caenorhabditis Genetics Center (CGC) whichis supported by the National Institutes of Health - Office of Research Infrastructure Programs (P40 OD010440) for strains, NemaGENETAG consortium for a *Mos1* insertion strain and M Sarov at TransgeneOme for fosmid reporters. This work was funded by the National Institutes of Health [R01 NS039996] and the Howard Hughes Medical Institute. MG was supported by EMBO and then HFSPO postdoctoral fellowships.

## Additional information

### Competing interests

OH: Reviewing editor, *eLife*. The other authors declare that no competing interests exist.

### Funding

| Funder | Grant reference number | Author |
| --- | --- | --- |
| EMBO | Postdoctoral fellowship | Marie Gendrel |
| Human Frontier Science Program | Postdoctoral fellowship | Marie Gendrel |
| National Institutes of Health | R01 NS039996 | Marie Gendrel<br>Oliver Hobert |
| Howard Hughes Medical Institute | | Marie Gendrel<br>Oliver Hobert |
| Howard Hughes Medical Institute | | Oliver Hobert |
| National Institute of Neurological Disorders and Stroke | 5R37NS039996-16 | Oliver Hobert |

The funders had no role in study design, data collection and interpretation, or the decision to submit the work for publication.

### Author contributions

MG, Conception and design, Acquisition of data, Analysis and interpretation of data, Drafting or revising the article; EGA, Acquisition of data, Analysis and interpretation of data; OH, Conception and design, Analysis and interpretation of data, Drafting or revising the article

### Author ORCIDs

Oliver Hobert, http://orcid.org/0000-0002-7634-2854

## Additional files

### Supplementary files

• Supplementary file 1. A list of all *C. elegans* neuron classes and their assigned neurotransmitter identities. Data from (*Chase and Koelle, 2007*; *Pereira et al., 2015*; *Serrano-Saiz et al., 2013*).

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
