## [Decision Letter]

Thank you for submitting your article "A cellular and regulatory map of the GABAergic nervous system of *C. elegans*" for consideration by *eLife*. Your article has been reviewed by three peer reviewers, and the evaluation has been overseen by Kang Shen as the Reviewing Editor and Eve Marder as the Senior Editor. Two of the three reviewers have agreed to reveal their identities: Piali Sengupta (Reviewer #1) and David M. Miller III (Reviewer #2).

The reviewers have discussed the reviews with one another and the Reviewing Editor has drafted this decision to help you prepare a revised submission.

All three reviewers thought that it is an excellent addition to the set of papers from the Hobert and other labs describing the classical neurotransmitter and peptidergic maps in *C. elegans*. Specific results that substantially expand our knowledge include the discovery of several additional GABA+ neurons that had not been previously reported and the novel finding of GABA+ glial cells. Transcription factors with necessary roles in the differentiation of specific GABA neurons are validated with genetic mutants, some of which were generated in this study. Together with the known anatomical connectivity, complete knowledge of the detailed transmitter expression patterns will be extremely useful in ultimately describing a functional connectivity map in this organism.

The reviewers did raise several points that are worth discussing:

1) Although the conserved enzyme UNC-25/GAD is necessary for GABA synthesis in most GABAergic neurons, a recent study determined that GABA can also be synthesized in the mammalian CNS from the polyamine putrescine by the successive action of the enzymes diamino oxidase (DAO) and alcohol dehydrogenase (ALDH) (Kim et al., 2015). Because this pathway is evolutionarily ancient with its initial discovery in plants, it offers a plausible explanation for the detection of GABA in the nematode neurons reported in this work that do not express UNC-25/GAD. Indeed, the putrescine pathway might explain the origin of GABA in neurons listed in Table 1 that express neither *unc-25* nor the GABA uptake transporter, SNF-11 (e.g., SMDD).

Recent work, has also reported GABA release from dopaminergic neurons that depends on a vesicular monamine transporter, VMAT2, rather than VGAT/UNC-47 (Tritsch, Ding, & Sabatini, 2012). The question then is whether VMAT/CAT-1 is expressed in GABA+ neurons (Table 1) that are negative for UNC-47 (e.g., AVA). This consideration is important because it suggests the possibility that this subset of neurons could actually modulate neuronal function by releasing GABA. At the very least, the authors should acknowledge these caveats in the Discussion.

Kim, J.-I., Ganesan, S., Luo, S.X., Wu, Y.-W., Park, E., Huang, E.J., Chen, L., and Ding, J.B. (2015). Aldehyde dehydrogenase 1a1 mediates a GABA synthesis pathway in midbrain dopaminergic neurons. Science 350, 102-106.

Tritsch, N.X., Ding, J.B., and Sabatini, B.L. (2012). Dopaminergic neurons inhibit striatal output through non-canonical release of GABA. Nature 490, 262-266.

2) The paper assumes all expression is functional. Thus, if GABA is present in a neuron, there must be some functional role for this presence. In contrast, without selective pressure against its presence, many non-functional expression of proteins exist. Indeed, evolution requires some of these redundancies since, for example it is difficult to co-evolve release and receptor in one step.

In particular, the presence of GABA-A receptors in neurons that do not have GABA inputs is not 'evidence' for spillover transmission. Similarly, the presence of GABA in neurons that do not have the GABA transporter is not 'evidence' for unconventional GABA release mechanisms. It seems more likely that these two are examples of non-functional expression that is not selected against.

As stated in the text and summarized in Table 2, the reporter for VGAT/UNC47 is present in many neurons that are not positive for GABA. A suggestion is made that in these neurons VGAT may transport an as yet unknown transmitter. This is a reasonable hypothesis, but it also suggests that in the absence of GAD, assuming that GABA immunoreactivity + VGAT necessarily indicates release of GABA may be an overstatement, without additional data that these neurons release GABA.

There are many transporters in the *C. elegans* genome and it is difficult to know with certainty that no other transporter can transport GABA. This should be a caveat of the assumption that neurons lacking SNF-11 generate their own GABA, despite being not positive for the GAD enzyme. While it is a possibility that this reporter is not complete, it is also possible that other transporters exist that can transport GABA. This should be acknowledged.

Our suggestion is to call those neurons "putative GABAergic neurons" in the title and throughout the manuscript, and to discuss these possibilities in the manuscript.

---

## [Author Response]

The reviewers did raise several points that are worth discussing:

*1) Although the conserved enzyme UNC-25/GAD is necessary for GABA synthesis in most GABAergic neurons, a recent study determined that GABA can also be synthesized in the mammalian CNS from the polyamine putrescine by the successive action of the enzymes diamino oxidase (DAO) and alcohol dehydrogenase (ALDH) (Kim et al., 2015). Because this pathway is evolutionarily ancient with its initial discovery in plants, it offers a plausible explanation for the detection of GABA in the nematode neurons reported in this work that do not express UNC-25/GAD. Indeed, the putrescine pathway might explain the origin of GABA in neurons listed in Table 1 that express neither unc-25 nor the GABA uptake transporter, SNF-11 (e.g., SMDD).*

We do not think that an alternative synthesis pathway plays any role in *C. elegans* because in *unc-25/GAD* mutants the anti-GABA staining is *completely eliminated* in *all* cells shown in Table 1. We emphasize this point in the Results section.

*Recent work, has also reported GABA release from dopaminergic neurons that depends on a vesicular monamine transporter, VMAT2, rather than VGAT/UNC-47 (Tritsch, Ding, & Sabatini, 2012). The question then is whether VMAT/CAT-1 is expressed in GABA+ neurons (Table 1) that are negative for UNC-47 (e.g., AVA). This consideration is important because it suggests the possibility that this subset of neurons could actually modulate neuronal function by releasing GABA. At the very least, the authors should acknowledge these caveats in the Discussion.*

Kim, J.-I., Ganesan, S., Luo, S.X., Wu, Y.-W., Park, E., Huang, E.J., Chen, L., and Ding, J.B. (2015). Aldehyde dehydrogenase 1a1 mediates a GABA synthesis pathway in midbrain dopaminergic neurons. Science 350, 102-106.

*Tritsch, N.X., Ding, J.B., and Sabatini, B.L. (2012). Dopaminergic neurons inhibit striatal output through non-canonical release of GABA. Nature 490, 262-266.*

This is an excellent point, thank you. The expression pattern of VMAT/CAT-1 is actually quite precisely known, based on antibody staining (Duerr, 1999) and fosmid reporters (our own unpublished data); expression is observed *exclusively* in all monoaminergic neurons (+ CAN neurons). *None* of the GABA(+); VGAT(-) cells show expression. However, this point is well taken and we now mention the possibility that other SLC-type transporters may package GABA into vesicles. Also, another unconventional release mechanism of GABA (from glia) – which we have only recently become aware of – depends on the ion channel bestrophin, of which there are several *C. elegans* homologs. We now also mention this possibility (subsection “VGAT expression suggests the existence of “recycling neurons” and “clearance neurons”).

2) The paper assumes all expression is functional. Thus, if GABA is present in a neuron, there must be some functional role for this presence. In contrast, without selective pressure against its presence, many non-functional expression of proteins exist. Indeed, evolution requires some of these redundancies since, for example it is difficult to co-evolve release and receptor in one step.

*In particular, the presence of GABA-A receptors in neurons that do not have GABA inputs is not 'evidence' for spillover transmission. Similarly, the presence of GABA in neurons that do not have the GABA transporter is not 'evidence' for unconventional GABA release mechanisms. It seems more likely that these two are examples of non-functional expression that is not selected against.*

We completely agree. In as far as GABA presence goes (particularly in the “unusual” neurons that don't have UNC-47), we do say “…whether and how they employ GABA for signaling is presently not clear” and “…may simply not engage in GABA signaling at all”. Regarding the GABA receptors and spill-over transmission, yes, we need to be more careful. We now state “These receptors may simply not function as GABA receptors or, more interestingly, these receptors may mediate GABA spillover transmission, a phenomenon observed in the vertebrate CNS as well as in *C. elegans*” and, in the Discussion, we reworded our conclusions to state: “…it remains to be experimentally tested whether these GABA receptors indeed engage in GABA spillover transmission.”

As stated in the text and summarized in Table 2, the reporter for VGAT/UNC47 is present in many neurons that are not positive for GABA. A suggestion is made that in these neurons VGAT may transport an as yet unknown transmitter. This is a reasonable hypothesis, but it also suggests that in the absence of GAD, assuming that GABA immunoreactivity + VGAT necessarily indicates release of GABA may be an overstatement, without additional data that these neurons release GABA.

Also completely agree. We reworded a sentence to now make clear that VGAT expression only “suggests” GABA release: “…suggesting that these neurons not only contain GABA but can also synaptically release it”. We actually now also qualified our statement that UNC-47 may act as a transporter for other transmitters (which is still possible): that’s because the protein that traffics UNC-47 to synaptic vesicles (UNC-46) is *not* expressed in most of the “new” UNC-47(+) neurons. UNC-47 may have non-vesicular functions in these cells. We now mention this in the subsection “2 VGAT expression suggests the existence of ‘recycling neurons’ and ‘clearance neurons’”.

*There are many transporters in the C. elegans genome and it is difficult to know with certainty that no other transporter can transport GABA. This should be a caveat of the assumption that neurons lacking SNF-11 generate their own GABA, despite being not positive for the GAD enzyme. While it is a possibility that this reporter is not complete, it is also possible that other transporters exist that can transport GABA. This should be acknowledged.*

Yes, good point, similar to the point above, on alternative VGATs. We have actually now also tested the transporter most closely related to SNF-11, called SNF-3, a validated betaine transporter. We do not see any effect on GABA staining in *snf-3* null mutants.

*Our suggestion is to call those neurons "putative GABAergic neurons" in the title and throughout the manuscript, and to discuss these possibilities in the manuscript.*

We beg to differ on this point. We completely agree that we do not know whether GABA is indeed employed by some neurons – but we do not believe that proof of GABA release/usage is a reasonable criterion to call a neuron GABAergic. By this criterion most GABAergic neurons in the vertebrate nervous system would need to be called “putative”. In fact, by this criterion even some of “classic” GABAergic neurons would have to be called “putative” (e.g. the known function of RIS does not require GABA). At this point, we consider it more useful to call any neuron “GABAergic” that stains positive for GABA.